# Improved Sample Complexity for Private Nonsmooth Nonconvex Optimization

## Abstract

We study differentially private (DP) optimization algorithms for stochastic and empirical objectives which are neither smooth nor convex, and propose methods that return a Goldstein-stationary point with sample complexity bounds that improve on existing works. We start by providing a single-pass $(\varepsilon, \delta)$-DP algorithm that returns an $(\alpha, \beta)$-stationary point as long as the dataset is of size $\widetilde{\Omega}\left(1/\alpha\beta^3 + d/\varepsilon\alpha\beta^2 + d^{3/4}/\varepsilon^{1/2}\alpha\beta^{5/2}\right)$, which is $\Omega(\sqrt{d})$ times smaller than the algorithm of Zhang et al. (2024) for this task, where $d$ is the dimension. We then provide a multi-pass polynomial time algorithm which further improves the sample complexity to $\widetilde{\Omega}\left(d/\beta^2 + d^{3/4}/\varepsilon\alpha^{1/2}\beta^{3/2}\right)$, by designing a sample efficient ERM algorithm, and proving that Goldstein-stationary points generalize from the empirical loss to the population loss.

## 1 Introduction

We consider optimization problems in which the loss function is stochastic or empirical, of the form

$$F(x) := \mathbb{E}_{\xi \sim \mathcal{P}} \left[ f(x; \xi) \right], \qquad \text{(stochastic)}$$

$$\widehat{F}^{\mathcal{D}}(x) := \frac{1}{n} \sum_{i=1}^{n} f(x; \xi_i), \qquad \text{(ERM)}$$

where $\mathcal{P}$ is the population distribution from which we sample a dataset $\mathcal{D} = (\xi_1, \ldots, \xi_n) \sim \mathcal{P}^n$, and the component functions $f(\,\cdot\,; \xi) : \mathbb{R}^d \to \mathbb{R}$ may be neither smooth nor convex. Such problems are ubiquitous throughout machine learning, where losses given by deep-learning based models give rise to highly nonsmooth nonconvex (NSNC) landscapes.

Due to its fundamental importance in modern machine learning, the field of nonconvex optimization has received substantial attention in recent years. Moving away from the classical regime of convex optimization, many works aimed at understanding the complexity of producing approximate-stationary points, namely with small gradient norm (Ghadimi & Lan, 2013; Fang et al., 2018; Carmon et al., 2020; Arjevani et al., 2023). As it turns out, without smoothness, it is impossible to directly minimize the gradient norm without suffering from an exponential-dimension dependent runtime in the worst case (Kornowski & Shamir, 2022). Nonetheless, a nuanced notion coined as Goldstein-stationarity (Goldstein, 1977), has been shown in recent years to enable favorable guarantees. Roughly speaking, a point $x \in \mathbb{R}^d$ is called an $(\alpha, \beta)$-Goldstein stationary point (or simply $(\alpha, \beta)$-stationary) if there exists a convex combination of gradients in the $\alpha$-ball around $x$ whose norm is at most $\beta$.[1] Following the groundbreaking work of Zhang et al. (2020), a surge of works study NSNC optimization through the lens of Goldstein stationarity, with associated finite-time guarantees (Davis et al., 2022; Lin et al., 2022; Cutkosky et al., 2023; Jordan et al., 2023; Kong & Lewis, 2023; Grimmer & Jia, 2023; Kornowski & Shamir, 2024; Tian & So, 2024).

In this work, we study NSNC optimization problems under the additional constraint of differential privacy (DP) (Dwork et al., 2006). With the ever-growing deployment of ML models in various domains, the privacy of the data on which models are trained is a major concern. Accordingly, DP

---

[1]Previous works typically use the notational convention $(\delta, \varepsilon)$-stationarity instead of $(\alpha, \beta)$, namely where $\delta$ is the radius (instead of $\alpha$) and $\varepsilon$ is the norm bound (instead of $\beta$). We depart from this notational convention in order to avoid confusion with the standard privacy notation of $(\varepsilon, \delta)$-DP.

optimization is an extremely well-studied problem, with a vast literature focusing on functions that are assumed to be either convex or smooth (Bassily et al., 2014; Wang et al., 2017; Bassily et al., 2019; Wang et al., 2019; Feldman et al., 2020; Gopi et al., 2022; Arora et al., 2023; Carmon et al., 2023; Liu et al., 2024). The fundamental investigation in this literature is the privacy-utility trade-off, that is, assessing the minimal dataset size $n$ (referred to as the sample complexity) required in order to optimize the loss up to some error, using a DP algorithm.

For NSNC DP optimization, Zhang et al. (2024) recently provided a zero-order algorithm, namely that utilizes only function value evaluations of $f(\cdot\,;\xi)$, which preforms a single pass over the dataset and returns an $(\alpha,\beta)$-stationary point of $F$ under $(\varepsilon,\delta)$-DP as long as

$$n = \widetilde{\Omega}\left(\frac{d}{\alpha\beta^3} + \frac{d^{3/2}}{\varepsilon\alpha\beta^2}\right). \tag{1}$$

To the best of our knowledge this is the only existing result in this realm.

## 1.1 OUR CONTRIBUTIONS.

In this paper, we provide new algorithms for NSNC DP optimization, which improve the previously best-known sample complexity for this task. For consistency with the previous result by Zhang et al. (2024), our algorithms will be zero-order, yet in Appendix C we provide first-order algorithms (i.e., gradient-based) with the same sample complexities, and better oracle complexity. Our contributions, summarized in Table 1, are as follows:

1. **Improved single-pass algorithm (Theorem 3.1):** We provide an $(\varepsilon,\delta)$-DP algorithm that pre-forms a single pass over that dataset, and returns an $(\alpha,\beta)$-stationary point as long as

$$n = \widetilde{\Omega}\left(\frac{1}{\alpha\beta^3} + \frac{d}{\varepsilon\alpha\beta^2} + \frac{d^{3/4}}{\varepsilon^{1/2}\alpha\beta^{5/2}}\right), \tag{2}$$

which is always at least $\Omega(\sqrt{d})$ times smaller than (1).[2] Notably, the "non-private" term $1/\alpha\beta^3$ is dimension-independent, as opposed to Eq. (1), which is the first result of this sort for NSNC DP optimization, and was erroneously claimed impossible by previous work (see Remark 3.2).

2. **Better multi-pass algorithm (Theorem 4.1):** In order to further improve the sample complexity, we move to consider ERM algorithms that go over the data multiple times (polynomially), which we will later argue generalize to the population loss. To that end, we provide an $(\varepsilon,\delta)$-DP ERM algorithm, that returns an $(\alpha,\beta)$-Goldstein stationary point of $\widehat{F}^{\mathcal{D}}$ as long as

$$n = \widetilde{\Omega}\left(\frac{d^{3/4}}{\varepsilon\alpha^{1/2}\beta^{3/2}}\right). \tag{3}$$

Notably, Eq. (3) substantially improves Eq. (2) (and thus, Eq. (1)) in parameter regimes of interest (small $\varepsilon,\alpha,\beta$, large $d$) with respect to the dimension and accuracy parameters, and in particular is the first algorithm to preform private ERM with sublinear dimension-dependent sample complexity for NSNC objectives.

In order to utilize our empirical algorithm for stochastic objectives, one must argue that Goldstein-stationarity generalizes from the ERM to the population. As no such argument is currently pointed out in the literature, we provide a result that ensures this:

- **Additional contribution: generalizing from ERM to population (Proposition 5.1).** We show that with high probability, any $(\alpha,\widehat{\beta})$-stationary point of $\widehat{F}^{\mathcal{D}}$ is an $(\alpha,\beta)$-stationary point of $F$, for $\beta = \widehat{\beta} + \widetilde{O}(\sqrt{d/n})$. Hence, the empirical guarantee Eq. (3) generalizes to stochastic losses with an additional $d/\beta^2$ additive term in $n$ (up to log terms).

---

[2]Note that $\frac{d^{3/4}}{\varepsilon^{1/2}\alpha\beta^{5/2}} \lesssim \frac{1}{\sqrt{d}}\left(\frac{d}{\alpha\beta^3} + \frac{d^{3/2}}{\varepsilon\alpha\beta^2}\right)$ by the AM-GM inequality

| Sample complexity summary | empirical | stochastic |
|---|---|---|
| (Zhang et al., 2024) (single-pass) | $\frac{d}{\alpha\beta^3} + \frac{d^{3/2}}{\varepsilon\alpha\beta^2}$ | |
| Theorem 3.1 (single-pass) | $\frac{1}{\alpha\beta^3} + \frac{d}{\varepsilon\alpha\beta^2} + \frac{d^{3/4}}{\varepsilon^{1/2}\alpha\beta^{5/2}}$ | |
| Theorem 4.1 (multi-pass) | $\frac{d^{3/4}}{\varepsilon\alpha^{1/2}\beta^{3/2}}$ | $\frac{d}{\beta^2} + \frac{d^{3/4}}{\varepsilon\alpha^{1/2}\beta^{3/2}}$ |

Table 1: Main results (ignoring dependence on Lipschitz constant, initialization, and log terms).

## 2 PRELIMINARIES

**Notation.** We denote by $\langle\cdot,\cdot\rangle, \|\cdot\|$ the standard Euclidean dot product and its induced norm. For $x \in \mathbb{R}^d$ and $\alpha > 0$, we denote by $\mathbb{B}(x,\alpha)$ the closed ball of radius $\alpha$ centered at $x$, and further denote $\mathbb{B}_\alpha := \mathbb{B}(0,\alpha)$. $\mathbb{S}^{d-1} \subset \mathbb{R}^d$ denotes the unit sphere. We make standard use of $O$-notation to hide absolute constants, $\widetilde{O}, \widetilde{\Omega}$ to hide poly-logarithmic factors, and also let $f \lesssim g$ denote $f = O(g)$.

**Nonsmooth optimization.** A function $h : \mathbb{R}^d \to \mathbb{R}$ is called $L$-Lipschitz if for all $x, y \in \mathbb{R}^d$ : $|h(x) - h(y)| \leq L\|x - y\|$. We call $h$ $H$-smooth, if $h$ is differentiable and $\nabla h$ is $H$-Lipschitz with respect to the Euclidean norm. For Lipschitz functions, the Clarke subgradient set (Clarke, 1990) can be defined as

$$\partial h(x) := \text{conv}\{g \, : \, g = \lim_{n\to\infty} \nabla h(x_n), \, x_n \to x\},$$

namely the convex hull of all limit points of $\nabla h(x_n)$ over sequences of differentiable points (which are a full Lebesgue-measure set by Rademacher's theorem), converging to $x$. For $\alpha \geq 0$, the Goldstein $\alpha$-subdifferential (Goldstein, 1977) is further defined as

$$\partial_\alpha h(x) := \text{conv}(\cup_{y\in\mathbb{B}(x,\alpha)}\partial h(y)),$$

and we denote the minimum-norm element of the Goldstein $\alpha$-subdifferential by

$$\overline{\partial}_\alpha h(x) := \arg\min_{g\in\partial_\alpha h(x)} \|g\|.$$

**Definition 2.1.** A point $x \in \mathbb{R}^d$ is called an $(\alpha, \beta)$-Goldstein stationary point of $h$ if $\left\|\overline{\partial}_\alpha h(x)\right\| \leq \beta$.

Throughout the paper we impose the following standard Lipschitz assumption:

**Assumption 2.2.** *For any $\xi$, $f(\cdot\,;\xi) : \mathbb{R}^d \to \mathbb{R}$ is $L$-Lipschitz (hence, so is $F$).*

**Randomized smoothing.** Given any function $h : \mathbb{R}^d \to \mathbb{R}$, we denote its randomized smoothing $h_\alpha(x) := \mathbb{E}_{y\sim\mathbb{B}_\alpha} h(x + y)$. We recall the following standard properties of randomized smoothing (Flaxman et al., 2005; Yousefian et al., 2012; Duchi et al., 2012; Shamir, 2017).

**Fact 2.3** (Randomized smoothing). *Suppose $h : \mathbb{R}^d \to \mathbb{R}$ is $L$-Lipschitz. Then*

- *$h_\alpha$ is $L$-Lipschitz.*

- *$|h_\alpha(x) - h(x)| \leq L\alpha$ for any $x \in \mathbb{R}^d$.*

- *$h_\alpha$ is $O(L\sqrt{d}/\alpha)$-smooth.*

- *$\nabla h_\alpha(x) = \mathbb{E}_{y\sim\mathbb{B}_\alpha}[\nabla h(x + y)] = \mathbb{E}_{y\sim\mathbb{S}^{d-1}}[\frac{d}{2\alpha}(h(x + \alpha y) - h(x - \alpha y))y].$*

The following result shows that in order to find a Goldstein-stationary point of a function, it suffices to find a Goldstein-stationary point of its randomized smoothing:

**Lemma 2.4** (Kornowski & Shamir, 2024, Lemma 4). *Any $(\alpha, \beta)$-stationary point of $h_\alpha$ is a $(2\alpha, \beta)$-stationary point of $h$.*

**Differential privacy.** Two datasets $\mathcal{D}, \mathcal{D}' \in \mathrm{supp}(\mathcal{P})^n$ are said to be neighboring if they differ in only one data point. A randomized algorithm $\mathcal{A} : \mathcal{Z}^n \to \mathcal{R}$ is called $(\varepsilon, \delta)$ differentially private (or $(\varepsilon, \delta)$-DP) for $\varepsilon, \delta > 0$ if for any two neighboring datasets $\mathcal{D}, \mathcal{D}'$ and measurable $E \subseteq \mathcal{R}$ in the algorithm's range, it holds that $\Pr[\mathcal{A}(\mathcal{D}) \in E] \leq e^\varepsilon \Pr[\mathcal{A}(\mathcal{D}') \in E] + \delta$ (Dwork et al., 2006).

Next, we recall the well-known tree mechanism given by Algorithm 1, and its associated guarantee presented below.

**Proposition 2.5** (Tree Mechanism Dwork et al., 2010; Chan et al., 2011; Zhang et al., 2024). *Let $\mathcal{Z}_1, \cdots, \mathcal{Z}_\Sigma$ be dataset spaces, and $\mathcal{X}$ be the state space. Let $\mathcal{M}_i : \mathcal{X}^{i-1} \times \mathcal{Z}_i \to \mathcal{X}$ be a sequence of algorithms for $i \in [\Sigma]$. Let $\mathrm{ALG} : \mathcal{Z}^{(1:\Sigma)} \to \mathcal{X}^\Sigma$ be the algorithm that given a dataset $Z_{1:\Sigma} \in \mathcal{Z}^{(1:\Sigma)}$, sequentially computes $X_i = \sum_{j=1}^i \mathcal{M}_i(X_{1:j-1}, Z_i) + \mathrm{TREE}(i)$ for $i \in [\Sigma]$, and then outputs $X_{1:\Sigma}$. Suppose for all $i \in [\Sigma]$, and neighboring $Z_{1:\Sigma}, Z'_{1:\Sigma} \in \mathcal{Z}^{(1:\Sigma)}, \|\mathcal{M}_i(X_{1:i-1}, Z_i) - \mathcal{M}_i(X_{1:i-1}, Z'_i)\| \leq s$ for all auxiliary inputs $X_{1:i-1} \in \mathcal{X}^{i-1}$. Then setting $\sigma = 4s\sqrt{\log \Sigma \log(1/\delta)}/\varepsilon$, Algorithm 1 is $(\varepsilon, \delta)$-DP. Furthermore, for all $t \in [\Sigma] : \mathbb{E}[\mathrm{TREE}(t)] = 0$ and $\mathbb{E}\|\mathrm{TREE}(t)\|^2 \lesssim d\log(\Sigma)\sigma^2$.*

---

**Algorithm 1** Tree Mechanism

1: **Input:** Noise parameter $\sigma$, sequence length $\Sigma$
2: Define $\mathcal{T} := \{(u, v) : u = j \cdot 2^{\ell-1} + 1, v = (j+1) \cdot 2^{\ell-1}, 1 \leq \ell \leq \log \Sigma, 0 \leq j \leq \Sigma/2^{\ell-1} - 1\}$
3: Sample and store $\zeta_{(u,v)} \sim \mathcal{N}(0, \sigma^2)$ for each $(u, v) \in \mathcal{T}$
4: **for** $t = 1, \cdots, \Sigma$ **do**
5:     Let $\mathrm{TREE}(t) \leftarrow \sum_{(u,v) \in \mathrm{NODE}(t)} \zeta_{(u,v)}$
6: **end for**
7: **Return:** $\mathrm{TREE}(t)$ for each $t \in [\Sigma]$
8:
9: **Function NODE:**
10: **Input:** index $t \in [\Sigma]$
11: Initialize $S = \{\}$ and $k = 0$
12: **for** $i = 1, \cdots, \lceil \log \Sigma \rceil$ while $k < t$ **do**
13:     Set $k' = k + 2^{\lceil \log \Sigma \rceil - i}$
14:     **if** $k' \leq k$ **then**
15:         $S \leftarrow S \cup \{(k+1, k')\}, k \leftarrow k'$
16:     **end if**
17: **end for**

---

## 2.1 BASE ALGORITHM: O2NC

Similar to Zhang et al. (2024), our general algorithm is based on the so-called "Online-to-Non-Convex conversion" (O2NC) of Cutkosky et al. (2023). We slightly modify previous proofs by disentangling the role of the variance of the gradient estimator vs. its second order moment, as follows:

**Proposition 2.6** (O2NC). *Suppose that $\mathcal{O}(\,\cdot\,)$ is a stochastic gradient oracle of some differentiable function $h : \mathbb{R}^d \to \mathbb{R}$, so that for all $z \in \mathbb{R}^d : \mathbb{E}\|\mathcal{O}(z) - \nabla h(z)\|^2 \leq G_0^2$ and $\mathbb{E}\|\mathcal{O}(z)\|^2 \leq G_1^2$. Then running Algorithm 2 with $\eta = \frac{D}{G_1\sqrt{M}}$, $MD \leq \alpha$, uses $T$ calls to $\mathcal{O}(\,\cdot\,)$, and satisfies*

$$\mathbb{E}\left\|\overline{\partial}_\alpha h(x^{\mathrm{out}})\right\| \leq \frac{h(x_0) - \inf h}{DT} + \frac{3G_1}{2\sqrt{M}} + G_0.$$

We provide a proof of Proposition 2.6 in Appendix B. Recalling that by Lemma 2.4 any $(\alpha, \beta)$-stationary point of $F_\alpha$ is a $(2\alpha, \beta)$-stationary point of $F$, we see that it is enough to design a private stochastic gradient oracle $\mathcal{O}$ of $\nabla F_\alpha$, while controlling its variance $G_0$ and second moment $G_1$. In the next sections, we show how to construct such private oracles and derive the corresponding guarantees through Proposition 2.6. As previously remarked, in the main text, our oracles will be based on zero-order queries of the component functions $f(\,\cdot\,, \xi)$, yet in Appendix C, we also show we can construct oracles with the same sample complexity using first-order queries with a lower oracle complexity.

---

**Algorithm 2** Nonsmooth Nonconvex Algorithm (based on O2NC (Cutkosky et al., 2023))

---

1: **Input:** Oracle $\mathcal{O} : \mathbb{R}^d \to \mathbb{R}^d$, initialization $x_0 \in \mathbb{R}^d$, clipping parameter $D > 0$, step size $\eta > 0$, averaging length $M \in \mathbb{N}$, iteration budget $T \in \mathbb{N}$.
2: **Initialize:** $\Delta_1 = \mathbf{0}$
3: **for** $t = 1, \ldots, T$ **do**
4:     Sample $s_t \sim \text{Unif}[0, 1]$
5:     $x_t = x_{t-1} + \Delta_t$
6:     $z_t = x_{t-1} + s_t \Delta_t$
7:     $\tilde{g}_t = \mathcal{O}(z_t)$
8:     $\Delta_{t+1} = \text{clip}_D (\Delta_t - \eta \tilde{g}_t)$                         $\triangleright \text{clip}_D(z) := \min\{1, \frac{D}{\|z\|}\} \cdot z$
9: **end for**
10: $K = \lfloor \frac{T}{M} \rfloor$
11: **for** $k = 1, \ldots, K$ **do**
12:     $\overline{x}_k = \frac{1}{M} \sum_{m=1}^{M} z_{(k-1)M+m}$
13: **end for**
14: Sample $x^{\text{out}} \sim \text{Unif}\{\overline{x}_1, \ldots, \overline{x}_K\}$
15: **Output:** $x^{\text{out}}$.

---

## 3 SINGLE-PASS ALGORITHM

---

**Algorithm 3** Single-pass instantiation of $\mathcal{O}(z_t)$ in Line 7 of Algorithm 2

---

1: **Input:** Current iterate $z_t$, time $t \in \mathbb{N}$, period length $\Sigma \in \mathbb{N}$, accuracy parameter $\alpha > 0$, batch sizes $B_1, B_2 \in \mathbb{N}$, gradient validation size $m \in \mathbb{N}$, noise level $\sigma > 0$.
2: **if** $t \bmod \Sigma = 1$ **then**
3:     Sample minibatch $S_t$ of size $B_1$ among unused samples
4:     **for** each sample $\xi_i \in S_t$ **do**
5:         Sample $y_1, \ldots, y_m \overset{iid}{\sim} \text{Unif}(\mathbb{S}^{d-1})$
6:         $\tilde{\nabla} f(z_t; \xi_i) = \frac{1}{m} \sum_{j \in [m]} \frac{d}{2\alpha} (f(z_t + \alpha y_j; \xi_i) - f(z_t - \alpha y_j; \xi_i)) y_j$
7:     **end for**
8:     $g_t = \frac{1}{B_1} \sum_{\xi_i \in S_t} \tilde{\nabla} f(z_t; \xi_i)$
9: **else**
10:     Sample minibatch $S_t$ of size $B_2$ among unused samples
11:     **for** each sample $\xi_i \in S_t$ **do**
12:         Sample $y_1, \ldots, y_{2m} \overset{iid}{\sim} \text{Unif}(\mathbb{B}_\alpha)$
13:         $\tilde{\nabla} f(z_t; \xi_i) = \frac{1}{m} \sum_{j \in [m]} \frac{d}{2\alpha} (f(z_t + \alpha y_j; \xi_i) - f(z_t - \alpha y_j; \xi_i)) y_j$
14:         $\tilde{\nabla} f(z_{t-1}; \xi_i) = \frac{1}{m} \sum_{j=m+1}^{2m} \frac{d}{2\alpha} (f(z_{t-1} + \alpha y_j; \xi_i) - f(z_{t-1} - \alpha y_j; \xi_i)) y_j$
15:     **end for**
16:     $g_t = g_{t-1} + \frac{1}{B_2} \sum_{\xi_i \in S_t} (\tilde{\nabla} f(z_t; \xi_i) - \tilde{\nabla} f(z_{t-1}; \xi_i))$
17: **end if**
18: **Return** $\tilde{g}_t = g_t + \text{TREE}(\sigma, \Sigma)(t \bmod \Sigma)$

---

In this section, we consider Algorithm 3, which provides an oracle to be used in Algorithm 2. Algorithm 3 is such that throughout $T$ calls, it uses each data point once, and hence, privacy is maintained with no need for composition. Before getting into the details, we will provide the main underlying idea. We consider the zero-order gradient estimator

$$\tilde{\nabla} f_\alpha(x; \xi) = \frac{1}{m} \sum_{j=1}^{m} \frac{d}{2\alpha} (f(x + \alpha y_j; \xi) - f(x - \alpha y_j; \xi)), \ \ y_1, \ldots, y_m \overset{iid}{\sim} \text{Unif}(\mathbb{S}^{d-1}), \quad (4)$$

which is an unbiased estimator of $\nabla f_\alpha(x; \xi)$, to which we then apply variance reduction. Zhang et al. (2024) considered the oracle above specifically with $m = d$, for which it is easy to bound the

sensitivity of this estimator over neighboring minibatches $\xi_{1:B}, \xi'_{1:B}$ of size $B$ by

$$\left\| \frac{1}{B} \sum_{i=1}^{B} \tilde{\nabla} f_\alpha(x; \xi_i) - \frac{1}{B} \sum_{i=1}^{B} \tilde{\nabla} f_\alpha(x; \xi'_i) \right\| \leq \frac{Ld}{B}. \tag{5}$$

Our key observation is that while this is indeed the worst-case sensitivity, we can get substantially lower sensitivity *with high probability*. For sufficiently large $m$, standard sub-Gaussian concentration properties ensure that $\tilde{\nabla} f_\alpha(x; \xi_i) \approx \nabla f_\alpha(x; \xi_i)$ with high probability, and hence under this event we show the sensitivity over a mini-batch can be decreased to an order of $\frac{L}{B}$. As this is a factor of $d$ smaller than Eq. (5), we can add significantly less noise in order to privatize, therefore leading to faster convergence to stationarity.

The main theorem in this section is the following:

**Theorem 3.1** (Single-pass algorithm). *Suppose $F(x_0) - \inf_x F(x) \leq \Phi$, that Assumption 2.2 holds, and let $\alpha, \beta, \delta, \varepsilon > 0$ such that $\alpha \leq \frac{\Phi}{L}$. Then setting $B_1 = \Sigma$, $B_2 = 1$, $M = \alpha/4D$, $m = \widetilde{O}(d^2 B_1^2 + \frac{d\alpha^2 B_2^2}{D^2})$, $\sigma = \widetilde{O}(\frac{L}{B_1 \varepsilon} + \frac{LD\sqrt{d}}{\alpha B_2 \varepsilon})$, $\Sigma = \widetilde{\Theta}((\frac{\alpha}{\varepsilon D})^{2/3})$, $D = \widetilde{\Theta}(\min\{(\frac{\Phi^2 \alpha}{L^2 T^2})^{1/3}, (\frac{\Phi \alpha \varepsilon}{dLT})^{1/2}, (\frac{\Phi^3 \alpha^2 \varepsilon}{d^{3/2} L^3 T^3})^{1/5}\})$, $T = \Theta(n)$, and running Algorithm 2 with Algorithm 3 as the oracle subroutine, is $(\varepsilon, \delta)$-DP. Furthermore, its output satisfies $\mathbb{E}\|\overline{\partial}_{2\alpha} F(x^{\mathrm{out}})\| \leq \beta$ as long as*

$$n = \widetilde{\Omega} \left( \frac{\Phi L^2}{\alpha \beta^3} + \frac{\Phi Ld}{\varepsilon \alpha \beta^2} + \frac{\Phi L^{3/2} d^{3/4}}{\varepsilon^{1/2} \alpha \beta^{5/2}} \right).$$

**Remark 3.2.** It is interesting to note that the "non-private" term $\Phi L^2/\alpha\beta^3$ in Theorem 3.1 is independent of the dimension $d$. Not only is this the first result of this sort, this was even (erroneously) claimed impossible by Zhang et al. (2024). The reason for this confusion is that while the optimal zero-order *oracle* complexity is $d/\alpha\beta^3$ (Kornowski & Shamir, 2024), and in particular must scale with the dimension (Duchi et al., 2015), the *sample* complexity might not.

In the rest of the section, we will present the basic properties of this oracle in terms of sensitivity (implying the privacy), variance and second moment. We will then plug these into Algorithm 2, which enables proving Theorem 3.1. Corresponding proofs are deferred to Appendix A.

**Lemma 3.3** (Sensitivity). *Consider the gradient oracle $\mathcal{O}(\cdot)$ in Algorithm 3 when acting on two neighboring minibatches $S_t$ and $S'_t$, and correspondingly producing $g_t$ and $g'_t$, respectively. If $t \bmod \Sigma = 1$, then it holds with probability at least $1 - \delta/2$ that*

$$\|g_t - g'_t\| \lesssim \frac{L}{B_1} + \frac{Ld\sqrt{\log(dB_1/\delta)}}{\sqrt{m}}.$$

*Otherwise, conditioned on $g_{t-1} = g'_{t-1}$, we have with probability at least $1 - \delta/2$:*

$$\|g_t - g'_t\| \lesssim \frac{L\sqrt{d}D}{\alpha B_2} + \frac{Ld\sqrt{\log(dB_1/\delta)}}{\sqrt{m}}.$$

With the sensitivity bound given by Lemma 3.3, we easily derive the privacy guarantee of our oracle from the Tree Mechanism (Proposition 2.5).

**Lemma 3.4** (Privacy). *Running Algorithm 3 with $m = O\left(\log(dB_2/\delta)(d^2 B_1^2 + \frac{d\alpha^2 B_2^2}{D^2})\right)$ and $\sigma = O\left(\frac{L\sqrt{\log(1/\delta)}}{B_1 \varepsilon} + \frac{LD\sqrt{d\log(1/\delta)}}{\alpha B_2 \varepsilon}\right)$ is $(\varepsilon, \delta)$-DP.*

We next analyze the variance and second moment of the gradient oracle.

**Lemma 3.5** (Variance). *In Algorithm 3, for all $t \in [T]$ it holds that*

$$\mathbb{E}\|\tilde{g}_t - \nabla F_\alpha(z_t)\|^2 \lesssim \frac{L^2}{B_1} + \frac{L^2 d^2}{B_1 m} + \frac{L^2 dD^2 \Sigma}{\alpha^2 B_2} + \sigma^2 d \log \Sigma + \frac{L^2 d^2 \Sigma}{mB_2},$$

$$\mathbb{E}\|\tilde{g}_t\|^2 \lesssim L^2 + \frac{L^2 d^2}{B_1 m} + \frac{L^2 dD^2 \Sigma}{\alpha^2 B_2} + \sigma^2 d \log \Sigma + \frac{L^2 d^2 \Sigma}{mB_2}.$$

Combining the ingredients that we have set up, we can derive Theorem 3.1.

*Proof of Theorem 3.1.* The privacy guarantee follows directly from Lemma 3.4, by noting that our parameter assignment implies $B_1 T/\Sigma + B_2 T = O(n)$, which allows letting $T = \Theta(n)$ while never re-using samples (hence no privacy composition is required). Therefore, it remains to show the utility bound. By applying Lemma 2.4 and Proposition 2.6, we get that

$$\mathbb{E}\|\overline{\partial}_{2\alpha}F(x^{\text{out}})\| \leq \mathbb{E}\|\overline{\partial}_\alpha F_\alpha(x^{\text{out}})\| \leq \frac{F_\alpha(x_0) - \inf F_\alpha}{DT} + \frac{3G_1}{2\sqrt{M}} + G_0$$

$$\leq \frac{2\Phi}{DT} + \frac{3G_1}{2\sqrt{M}} + G_0, \tag{6}$$

where the last inequality used the fact that Assumption 2.2 and Fact 2.3 together imply that $F_\alpha(x_0) - \inf F_\alpha \leq F(x_0) - \inf F + L\alpha \leq \Phi + L\alpha \leq 2\Phi$. Under our parameter assignment, Lemma 3.5 yields

$$G_1 \lesssim G_0 + L, \tag{7}$$

which plugged into Eq. (6) gives

$$\mathbb{E}\|\overline{\partial}_{2\alpha}F(x^{\text{out}})\| = O\left(\frac{\Phi}{DT} + \frac{L}{\sqrt{M}} + G_0\right). \tag{8}$$

Moreover, under our parameter assignment Lemma 3.5 also gives the bound

$$G_0 \lesssim \frac{L}{B_1} + \frac{LD\sqrt{d\Sigma}}{\alpha B_2} + \sigma\sqrt{d\log\Sigma} = \widetilde{O}\left(\frac{LDd^{1/2}\Sigma^{1/2}}{\alpha} + \frac{Ld^{1/2}}{\Sigma\varepsilon} + \frac{LDd}{\alpha\varepsilon}\right), \tag{9}$$

which propagated into Eq. (8) and recalling that $M = \Theta(\alpha/D)$ shows that

$$\mathbb{E}\|\overline{\partial}_{2\alpha}F(x^{\text{out}})\| = \widetilde{O}\left(\frac{\Phi}{DT} + \frac{LD^{1/2}}{\alpha^{1/2}} + \frac{LDd^{1/2}\Sigma^{1/2}}{\alpha} + \frac{Ld^{1/2}}{\Sigma\varepsilon} + \frac{LDd}{\alpha\varepsilon}\right).$$

Plugging our assignments of $\Sigma$ and $D$, and recalling that $n = \Theta(T)$, a straightforward calculation simplifies the bound above to

$$\mathbb{E}\|\overline{\partial}_{2\alpha}F(x^{\text{out}})\| = \widetilde{O}\left(\left(\frac{\Phi L^2}{T\alpha}\right)^{1/3} + \left(\frac{\Phi dL}{T\alpha\varepsilon}\right)^{1/2} + \left(\frac{\Phi^2 L^3 d^{3/2}}{T^2\alpha^2\varepsilon}\right)^{1/5}\right)$$

$$= \widetilde{O}\left(\left(\frac{\Phi L^2}{n\alpha}\right)^{1/3} + \left(\frac{\Phi dL}{n\alpha\varepsilon}\right)^{1/2} + \left(\frac{\Phi^2 L^3 d^{3/2}}{n^2\alpha^2\varepsilon}\right)^{1/5}\right). \tag{10}$$

Bounding the latter by $\beta$ and solving for $n$ completes the proof.

$\square$

## 4 MULTI-PASS ALGORITHM

In this section, we consider a different oracle construction given by Algorithm 4, to be used in Algorithm 2. The main difference from the previous section is that this oracle reuses data points a polynomial number of times, and therefore cannot *directly* guarantee generalization to the stochastic objective. Instead, in this section we analyze the empirical objective $\widehat{F}^{\mathcal{D}}(x) := \frac{1}{n}\sum_{i=1}^n f(x;\xi_i)$. After establishing ERM results, in Section 5, we will show that any empirical Goldstein-stationarity guarantee generalizes to the population loss.

Similarly to the single-pass oracle (Algorithm 3), we use randomized smoothing and variance reduction. A difference in the oracle construction is that we replace the tree mechanism with the Gaussian mechanism and apply advanced composition for the privacy analysis (since now samples are reused). The main theorem for this section is the following:

---

**Algorithm 4** Multi-pass instantiation of $\mathcal{O}(z_t)$ in Line 7 of Algorithm 2

---

1: **Input:** Current iterate $z_t$, time $t \in \mathbb{N}$, period length $\Sigma \in \mathbb{N}$, accuracy parameter $\alpha > 0$, gradient validation size $m \in \mathbb{N}$, noise levels $\sigma_1$, $\sigma_2 > 0$.
2: **if** $t \bmod \Sigma = 1$ **then**
3:     **for** each sample $\xi_i \in \mathcal{D}$ **do**
4:         Sample $y_1, \ldots, y_m \overset{iid}{\sim} \mathrm{Unif}(\mathbb{S}^{d-1})$
5:         $\tilde{\nabla} f(z_t; \xi_i) = \frac{1}{m} \sum_{j \in [m]} \frac{d}{2\alpha}(f(z_t + \alpha y_j; \xi_i) - f(z_t - \alpha y_j; \xi_i))y_j$
6:     **end for**
7:     $g_t = \frac{1}{n} \sum_{\xi_i \in \mathcal{D}} \tilde{\nabla} f(z_t; \xi_i)$
8:     **Return:** $\tilde{g}_t = g_t + \chi_t$, where $\chi_t \sim \mathcal{N}(0, \sigma_1^2 I_d)$
9: **else**
10:     **for** each sample $\xi_i \in \mathcal{D}$ **do**
11:         Sample $y_1, \ldots, y_{2m} \overset{iid}{\sim} \mathrm{Unif}(\mathbb{B}_\alpha)$
12:         $\tilde{\nabla} f(z_t; \xi_i) = \frac{1}{m} \sum_{j=1}^{m} \frac{d}{2\alpha}(f(z_t + \alpha y_j; \xi_i) - f(z_t - \alpha y_j; \xi_i))y_j$
13:         $\tilde{\nabla} f(z_{t-1}; \xi_i) = \frac{1}{m} \sum_{j=m+1}^{2m} \frac{d}{2\alpha}(f(z_{t-1} + \alpha y_j; \xi_i) - f(z_{t-1} - \alpha y_j; \xi_i))y_j$
14:     **end for**
15:     $g_t = \tilde{g}_{t-1} + \frac{1}{n} \sum_{\xi_i \in \mathcal{D}}(\tilde{\nabla} f(z_t; \xi_i) - \tilde{\nabla} f(z_{t-1}; \xi_i))$
16:     **Return:** $\tilde{g}_t = g_t + \chi_t$, where $\chi_t \sim \mathcal{N}(0, \sigma_2^2 I_d)$.
17: **end if**

---

**Theorem 4.1** (Multi-pass ERM). *Suppose $\widehat{F}^{\mathcal{D}}(x_0) - \inf_x \widehat{F}^{\mathcal{D}}(x) \leq \Phi$, Assumption 2.2 holds, and let $\alpha, \beta, \delta, \varepsilon > 0$ such that $\alpha \leq \frac{\Phi}{L}$. Then setting $m = \frac{L^2 d\Sigma}{n\sigma_1^2} + \frac{L^2 d}{n\sigma_2^2}$, $\sigma_1 = O(\frac{L\sqrt{T \log(1/\delta)/\Sigma}}{n\varepsilon})$, $\sigma_2 = O(\frac{LD\sqrt{Td \log(1/\delta)}}{\alpha n\varepsilon})$, $\Sigma = \tilde{\Theta}(\frac{\alpha}{D\sqrt{d}})$, $D = \tilde{\Theta}(\frac{\alpha^2 \beta^2}{L^2})$, $T = \tilde{\Theta}(\frac{\Phi L^2}{\alpha^2 \beta^3})$, and running Algorithm 2 with Algorithm 4 as the oracle subroutine is $(\varepsilon, \delta)$-DP. Furthermore, its output satisfies $\mathbb{E}\|\overline{\partial}_{2\alpha} \widehat{F}^{\mathcal{D}}(x^{\mathrm{out}})\| \leq \beta$ as long as*

$$n = \widetilde{\Omega}\left(\frac{\sqrt{\Phi}Ld^{3/4}}{\varepsilon\alpha^{1/2}\beta^{3/2}}\right).$$

**Remark 4.2.** As we will show in Section 5, Theorem 4.1 also provides the same population guarantee for $\|\overline{\partial}_{2\alpha} F(x^{\mathrm{out}})\|$ with an additional $L^2 d/\beta^2$ term (up to log factors) to the sample complexity.

To prove Theorem 4.1, we analyze the properties of the oracle given by Algorithm 4. The sensitivity of $g_t$ in Algorithm 4 directly follows from Lemma 3.3.[3] By the standard composition results of the Gaussian mechanism (e.g., Mironov 2017), we have the following privacy guarantee:

**Lemma 4.3** (Privacy). *Calling Algorithm 4 $T$ times with $m = \frac{L^2 d\Sigma}{n\sigma_1^2} + \frac{L^2 d}{n\sigma_2^2}$, $\sigma_1 = O(\frac{L\sqrt{T \log(1/\delta)/\Sigma}}{n\varepsilon})$ and $\sigma_2 = O(\frac{LD\sqrt{Td \log(1/\delta)}}{\alpha n\varepsilon})$ is $(\varepsilon, \delta)$-DP.*

In terms of the oracle's variance, we show:

**Lemma 4.4** (Variance). *In Algorithm 4, for any $t \in [T]$, we have*

$$\mathbb{E}\|\tilde{g}_t - \nabla F_\alpha^{\mathcal{D}}(z_t)\|^2 \lesssim \frac{L^2 d^2 \Sigma}{mn} + \sigma_1^2 d + \sigma_2^2 d\Sigma,$$

$$\mathbb{E}\|\tilde{g}_t\|^2 \lesssim L^2 + \frac{L^2 d^2 \Sigma}{mn} + \sigma_1^2 d + \sigma_2^2 d\Sigma.$$

The proof of Theorem 4.1, which we defer to Appendix A, is a combination of the two previous lemmas and Proposition 2.6.

---

[3]In this section we use full-batch size for simplicity, but using smaller batches (of arbitrary size) and applying privacy amplification by subsampling, yields the same results up to constants.

## 5 EMPIRICAL TO POPULATION GOLDSTEIN-STATIONARITY

In this section, we provide a generalization result, showing that our ERM algorithm from the previous section also guarantees Goldstein-stationarity in terms of the population loss. We prove the following more general statement:

**Proposition 5.1.** *Under Assumption 2.2, suppose $\mathcal{D} \sim \mathcal{P}^n$, and consider running an algorithm on $\widehat{F}^{\mathcal{D}}$ whose (possibly randomized) output $x^{\mathrm{out}} \in \mathcal{X} \subset \mathbb{R}^d$ is supported over a set $\mathcal{X}$ of diameter $\leq R$. Then with probability at least $1 - \zeta: \|\overline{\partial}_\alpha F(x^{\mathrm{out}})\| \leq \|\overline{\partial}_\alpha \widehat{F}^{\mathcal{D}}(x^{\mathrm{out}})\| + \widetilde{O}\left(L\sqrt{d\log(R/\zeta)/n}\right)$.*

We remark that in all algorithms of interest, the output is known to lie in some predefined set, such as a sufficiently large ball around the initialization. As long as the diameter $R$ is polynomial in the problem parameters, the $\log(R)$ in the result above is therefore negligible. For instance, Algorithm 2 is easily verified to output a point $x^{\mathrm{out}} \in \mathbb{B}(x_0, DT)$ (since $\|x_{t+1} - x_t\| \leq D$). Hence, in our use case, Proposition 5.1 ensures $\|\overline{\partial}_{2\alpha} F(x^{\mathrm{out}})\| \leq \|\overline{\partial}_{2\alpha} \widehat{F}^{\mathcal{D}}(x^{\mathrm{out}})\| + \beta$ for $n = \widetilde{O}(d/\beta^2)$.

## 6 DISCUSSION

In this paper, we studied nonsmooth nonconvex optimization, and proposed differentially private algorithms for this task which return Goldstein-stationary points, improving the previously known sample complexity for this task.

Our single-pass algorithm reduces the sample complexity by at least a $\Omega(\sqrt{d})$ factor (and sometimes more, depending on the parameter regime of interest), compared to the previous such result by Zhang et al. (2024). Furthermore, our result has a dimension-independent "non-private" term, which was previously claimed impossible. Moreover, we propose a multi-pass algorithm which preforms sample-efficient ERM, and show that it further generalizes to the population.

It is interesting to note that our guarantees are in terms of so-called "approximate" $(\varepsilon, \delta)$-DP, whereas Zhang et al. (2024) derive a Rényi-DP guarantee (Mironov, 2017). This is in fact inherent to our techniques, since we condition on a highly probable event in order to substantially decrease the effective sensitivity of our gradient estimators. Further examining this potential gap in terms of sample complexity between approximate- and Rényi-DP for nonsmooth nonconvex optimization is an interesting direction for future research.

Another important problem that remains open is establishing tight lower bounds for DP nonconvex optimization and perhaps further improving the sample complexities obtained in this paper. We note that the current upper and lower bounds do not fully match even in the smooth setting. In Appendix D, we provide evidence that our upper bound can be further improved, by proposing a computationally-*inefficient* algorithm, which converges to a relaxed notion of stationarity, using even fewer samples than the algorithms we presented in this work.

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

# A PROOFS

## A.1 PROOFS FROM SECTION 3

*Proof of Lemma 3.3.* Note that for any $y \in \text{Unif}(\mathbb{S}^{d-1}) : \|\frac{d}{2\alpha}(f(z + \alpha y; \xi) - f(z - \alpha y; \xi))y\| \leq Ld$ due to the Lipschitz assumption. Hence, for any $\xi \in S_t$, by a standard sub-Gaussian bound (Theorem E.2) we have

$$\Pr\left[\|\tilde{\nabla}f(z_t; \xi) - \nabla f_\alpha(z_t; \xi)\| \leq \frac{Ld\sqrt{\log(8dB_1/\delta)}}{\sqrt{m}}\right] \geq 1 - \delta/8B_1. \tag{11}$$

If $t \bmod \Sigma = 1$, then

$$\|g_t - g_t'\| = \left\|\frac{1}{B_1}(\sum_{\xi \in S_t} \tilde{\nabla}f(z_t; \xi) - \sum_{\xi' \in S_t'} \tilde{\nabla}f(z_t; \xi'))\right\|$$

$$\leq \left\|\frac{1}{B_1}(\sum_{\xi \in S_t} \tilde{\nabla}f(z_t; \xi) - \nabla f_\alpha(z_t; \xi))\right\| + \left\|\frac{1}{B_1}(\sum_{\xi \in S_t} \nabla f_\alpha(z_t; \xi) - \sum_{\xi' \in S_t'} \nabla f_\alpha(z_t; \xi'))\right\|$$

$$+ \left\|\frac{1}{B_1}(\sum_{\xi' \in S_t'} \tilde{\nabla}f(z_t; \xi') - \nabla f_\alpha(z_t; \xi'))\right\|.$$

Further note that $\|\frac{1}{B_1}(\sum_{\xi \in S_t} \nabla f_\alpha(z_t; \xi) - \sum_{\xi' \in S_t'} \nabla f_\alpha(z_t; \xi'))\| \leq 2L/B_1$, hence by Equation Eq. (11) and the union bound,

$$\Pr\left[\left\|\frac{1}{B_1}(\sum_{\xi \in S_t} \tilde{\nabla}f(z_t; \xi) - \sum_{\xi' \in S_t'} \tilde{\nabla}f(z_t; \xi'))\right\| \geq \frac{Ld\sqrt{\log(8dB_1/\delta)}}{\sqrt{m}}\right] \leq 1 - \delta/8,$$

which proves the claim in the case when $t \bmod \Sigma = 1$. The other case follows from the same argument. $\square$

*Proof of Lemma 3.4.* By Lemma 3.3 and our assignment of $m$, we know that with probability at least $1 - \delta/2$, the sensitivity of all $t$ is bounded by $O(\frac{L}{B_1} + \frac{L\sqrt{d}D}{\alpha B_2})$, namely for all $t$ :

$$\|g_t - g_t'\| \lesssim \frac{L}{B_1} + \frac{L\sqrt{d}D}{\alpha B_2}.$$

Then the privacy guarantee follows from the Tree Mechanism (Proposition 2.5). $\square$

*Proof of Lemma 3.5.* First, note that by Proposition 2.5 and the facts that $\mathbb{E}[g_t] = \nabla F_\alpha(z_t)$ and $\|\nabla F_\alpha(z_t)\| \leq L$, we get

$$\mathbb{E}\|\tilde{g}_t\|^2 \lesssim \mathbb{E}\|g_t\|^2 + d\sigma^2 \log \Sigma \lesssim \mathbb{E}\|g_t - \nabla F_\alpha(z_t)\|^2 + L^2 + d\sigma^2 \log \Sigma,$$

and also

$$\mathbb{E}\|\tilde{g}_t - \nabla F_\alpha(z_t)\|^2 \lesssim \mathbb{E}\|\tilde{g}_t - g_t\|^2 + \mathbb{E}\|g_t - \nabla F_\alpha(z_t)\|^2 \lesssim d\sigma^2 \log \Sigma + \mathbb{E}\|g_t - \nabla F_\alpha(z_t)\|^2.$$

Therefore, we see that in order to obtain both claimed bounds, it suffices to bound $\mathbb{E}\|g_t - \nabla F_\alpha(z_t)\|^2$. To that end, denote by $t_0 \leq t$ the largest integer such that $t_0 \bmod \Sigma = 1$, and note that $t - t_0 < \Sigma$. Further denote $\Delta_j := g_j - g_{j-1}$. Then we have

$$\mathbb{E}\|g_t - \nabla F_\alpha(z_t)\|^2 = \mathbb{E}\left\|g_{t_0} + \sum_{j=t_0+1}^{t} \Delta_j - \left(\sum_{j=t_0+1}^{t} (\nabla F_\alpha(z_j) - \nabla F_\alpha(z_{j-1})) + \nabla F_\alpha(z_{t_0})\right)\right\|^2$$

$$= \underbrace{\mathbb{E}\|g_{t_0} - \nabla F_\alpha(z_{t_0})\|^2}_{(I)} + \sum_{j=t_0}^{t} \underbrace{\mathbb{E}\|\Delta_j - (\nabla F_\alpha(z_j) - \nabla F_\alpha(z_{j-1}))\|^2}_{(II)}, \tag{12}$$

where the last equality is due to the cross terms having zero mean. We further see that

$$(I) \lesssim \mathbb{E}\left\|g_{t_0} - \frac{1}{B_1}\sum_{\xi \in S_{t_0}} \nabla f_\alpha(z_{t_0}; \xi_i)\right\|^2 + \mathbb{E}\left\|\frac{1}{B_1}\sum_{\xi \in S_{t_0}} \nabla f_\alpha(z_{t_0}; \xi_i) - \nabla F_\alpha(z_{t_0})\right\|^2$$

$$\lesssim \frac{L^2 d^2}{B_1 m} + \frac{L^2}{B_1}, \tag{13}$$

as well as

$$(II) = \mathbb{E}\left\|\frac{1}{B_2}\sum_{\xi \in S_t} (\tilde{\nabla} f(z_j; \xi) - \tilde{\nabla} f(z_{j-1}; \xi)) - (\nabla F_\alpha(z_j) - \nabla F_\alpha(z_{j-1}))\right\|^2$$

$$= \frac{1}{B_2^2}\sum_{\xi \in S_t} \mathbb{E}\|(\tilde{\nabla} f(z_j; \xi) - \tilde{\nabla} f(z_{j-1}; \xi)) - (\nabla F_\alpha(z_j) - \nabla F_\alpha(z_{j-1}))\|^2$$

$$\lesssim \frac{1}{B_2^2}\sum_{\xi \in S_t} \Big(\mathbb{E}\|\tilde{\nabla} f(z_j; \xi) - \nabla f_\alpha(z_j; \xi)\|^2 + \mathbb{E}\|\tilde{\nabla} f(z_{j-1}; \xi) - \nabla f_\alpha(z_{j-1}; \xi)\|^2$$

$$+ \mathbb{E}\|(\nabla f_\alpha(z_j; \xi) - \nabla f_\alpha(z_{j-1}; \xi)) - (\nabla F_\alpha(z_j) - \nabla F_\alpha(z_{j-1}))\|^2\Big)$$

$$\lesssim \frac{L^2 d^2}{m B_2} + \frac{d L^2 D^2}{\alpha^2 B_2}. \tag{14}$$

Plugging Eq. (13) and Eq. (14) into Eq. (12) and recalling that $t - t_0 < \Sigma$ completes the proof. $\square$

## A.2 Proofs from Section 4

*Proof of Lemma 4.4.* First, it suffices to prove the first bound, as

$$\mathbb{E}\|\tilde{g}_t\|^2 \lesssim \mathbb{E}\|\tilde{g}_t - \nabla F_\alpha^{\mathcal{D}}(z_t)\|^2 + \mathbb{E}\|\nabla F_\alpha^{\mathcal{D}}(z_t)\|^2 \leq \mathbb{E}\|\tilde{g}_t - \nabla F_\alpha^{\mathcal{D}}(z_t)\|^2 + L^2.$$

To that end, let $t_0 \leq t$ be the largest integer such that $t_0 \bmod \Sigma \equiv 1$, and note that $t - t_0 < \Sigma$. Define $\Delta_j := \frac{1}{n}\sum_{\xi \in \mathcal{D}}(\tilde{\nabla} f(z_j; \xi) - \tilde{\nabla} f(z_{j-1}; \xi))$. It holds that

$$\mathbb{E}\|\tilde{g}_t - \nabla F_\alpha^{\mathcal{D}}(z_t)\|^2 \leq \underbrace{\mathbb{E}\|g_{t_0} - \nabla F_\alpha^{\mathcal{D}}(z_{t_0})\|^2}_{(I)} + \sum_{j=t_0}^{t}\underbrace{\mathbb{E}\|\Delta_j - (\nabla F_\alpha^{\mathcal{D}}(z_j) - \nabla F_\alpha^{\mathcal{D}}(z_{j-1}))\|^2}_{(II)} + \underbrace{\sum_{j=t_0}^{t}\mathbb{E}\|\chi_j\|^2}_{(III)}.$$

Similar to the proof of Lemma 3.5, we have that

$$(I) = \mathbb{E}\left\|g_{t_0} - \frac{1}{n}\sum_{\xi \in \mathcal{D}} \nabla f_\alpha(z_{t_0}; \xi_i)\right\|^2 \lesssim \frac{L^2 d^2}{nm},$$

$$(II) = \frac{1}{n^2}\mathbb{E}\|\sum_{\zeta \in \mathcal{D}}(\tilde{\nabla} f(z_j; \xi) - \tilde{\nabla} f(z_{j-1}; \xi)) - (\nabla \widehat{F}_\alpha^{\mathcal{D}}(z_j) - \nabla \widehat{F}_\alpha^{\mathcal{D}}(z_{j-1}))\|^2$$

$$\lesssim \frac{1}{n^2}\sum_{\xi \in \mathcal{D}} \Big(\mathbb{E}\|\tilde{\nabla} f(z_j; \xi) - \nabla f_\alpha(z_j; \xi)\|^2 + \mathbb{E}\|\tilde{\nabla} f(z_{j-1}; \xi) - \nabla f_\alpha(z_{j-1}; \xi)\|^2\Big)$$

$$\lesssim \frac{L^2 d^2}{mn},$$

$$(III) \leq \sigma_1^2 d + \sigma_2^2(\Sigma - 1),$$

overall completing the proof.

$\square$

*Proof of Theorem 4.1.* Setting $m = \frac{L^2 d\Sigma}{n\sigma_1^2} + \frac{L^2 d}{n\sigma_2^2}$, $\sigma_1 = O(\frac{L\sqrt{T\log(1/\delta)/\Sigma}}{n\varepsilon})$ and $\sigma_2 = O(\frac{LD\sqrt{Td\log(1/\delta)}}{\alpha n\varepsilon})$, the privacy guarantee follows from Lemma 4.3. Moreover, by our parameter settings, we have

$$G_0^2 := \mathbb{E}\|\tilde{g}_t - \nabla F_\alpha^{\mathcal{D}}(z_t)\|^2 \lesssim \frac{L^2 dT\log(1/\delta)/\Sigma}{n^2\varepsilon^2} + \frac{L^2 D^2 T d^2\Sigma\log(1/\delta)}{\alpha^2 n^2\varepsilon^2},$$

$$G_1^2 := \mathbb{E}\|\tilde{g}_t\|^2 \lesssim L^2 + \frac{L^2 dT\log(1/\delta)/\Sigma}{n^2\varepsilon^2} + \frac{L^2 D^2 T d^2\Sigma\log(1/\delta)}{\alpha^2 n^2\varepsilon^2}.$$

Therefore, setting $\Sigma = \widetilde{\Theta}(\frac{\alpha}{D\sqrt{d}})$, we see that $G_0 = \widetilde{O}(\frac{L\sqrt{DT}d^{3/4}}{n\varepsilon\sqrt{\alpha}})$ and $G_1 \lesssim L + G_0$. By Proposition 2.6, we also know that

$$\mathbb{E}\|\overline{\partial}_{2\alpha}\widehat{F}^{\mathcal{D}}(x^{\mathrm{out}})\| \leq \mathbb{E}\|\overline{\partial}_\alpha\widehat{F}_\alpha^{\mathcal{D}}(x^{\mathrm{out}})\| \leq \frac{F_\alpha(x_0) - \inf F_\alpha}{DT} + \frac{3G_1}{2\sqrt{M}} + G_0$$

$$\leq \frac{2\Phi}{DT} + \frac{3G_1}{2\sqrt{M}} + G_0.$$

Recalling that $M = \Theta(\alpha/D)$ and setting $D = \tilde{\Theta}(\frac{\alpha^2\beta^2}{L^2})$, $T = \tilde{\Theta}(\frac{\Phi L^2}{\alpha^2\beta^3})$, we have

$$\mathbb{E}\|\overline{\partial}_\alpha\widehat{F}_\alpha^{\mathcal{D}}(x^{\mathrm{out}})\| = \widetilde{O}\left(\frac{\Phi}{DT} + \frac{L\sqrt{D}}{\sqrt{\alpha}} + \frac{L\sqrt{DT}d^{3/4}}{n\varepsilon\sqrt{\alpha}}\right)$$

$$= \frac{\beta}{2} + \widetilde{O}\left(\frac{Ld^{3/4}\sqrt{\Phi}}{n\varepsilon\sqrt{\alpha}\beta}\right).$$

The latter is bounded by $\beta$ for $n = \widetilde{\Omega}\left(\frac{L\sqrt{\Phi}d^{3/4}}{\varepsilon\alpha^{1/2}\beta^{3/2}}\right)$, hence completing the proof.

$\square$

### A.3 PROOFS FROM SECTION 5

*Proof of Proposition 5.1.* Applying a gradient uniform convergence bound for Lipschitz objectives over a bounded domain (Mei et al., 2018, Theorem 1), shows that with probability at least $1 - \zeta$, for any differentiable $x \in \mathcal{X}$:

$$\left\|\nabla\widehat{F}^{\mathcal{D}}(x) - \nabla F(x)\right\| = \widetilde{O}\left(L\sqrt{\frac{d\log(R/\zeta)}{n}}\right). \tag{15}$$

Therefore, given any $x \in \mathcal{X}$, let $y_1, \ldots, y_k \in \mathbb{B}(x, \alpha)$ be points satisfying $\overline{\partial}_\alpha\widehat{F}^{\mathcal{D}}(x) = \sum_{i=1}^k \lambda_i\nabla\widehat{F}^{\mathcal{D}}(y_i)$ for coefficients $(\lambda_i)_{i=1}^k \geq 0, \sum_{i=1}^k \lambda_i = 1$ — note that such points exist by definition of the Goldstein subdifferential. Noting that $\sum_{i=1}^k \lambda_i\nabla F(y_i) \in \partial_\alpha F(x)$, and recalling that $\overline{\partial}_\alpha F(x)$ is the minimal norm element of $\partial_\alpha F(x)$, we get that

$$\left\|\overline{\partial}_\alpha F(x)\right\| \leq \left\|\sum_{i=1}^k \lambda_i\nabla F(y_i)\right\| = \left\|\sum_{i=1}^k \lambda_i(\nabla\widehat{F}^{\mathcal{D}}(y_i) + \upsilon_i)\right\| = (\star)$$

where $\upsilon_i := \nabla F(y_i) - \nabla\widehat{F}^{\mathcal{D}}(y_i)$ satisfy $\|\upsilon_i\| = \widetilde{O}\left(L\sqrt{\frac{d\log(R/\zeta)}{n}}\right)$ for all $i \in [k]$ by Eq. (15). Hence

$$(\star) \leq \left\|\sum_{i=1}^k \lambda_i\nabla\widehat{F}^{\mathcal{D}}(y_i)\right\| + \left\|\sum_{i=1}^k \lambda_i\upsilon_i\right\|$$

$$\leq \left\|\overline{\partial}_\alpha\widehat{F}^{\mathcal{D}}(x)\right\| + \sum_{i=1}^k \lambda_i\|\upsilon_i\|$$

$$\leq \left\|\overline{\partial}_\alpha\widehat{F}^{\mathcal{D}}(x)\right\| + \widetilde{O}\left(L\sqrt{\frac{d\log(R/\zeta)}{n}}\right).$$

$\square$

# B    PROOF OF PROPOSITION 2.6 (O2NC)

We start by noting that the update rule for $\Delta_t$ which is given by

$$\Delta_{t+1} = \text{clip}_D \left( \Delta_t - \eta \tilde{g}_t \right) = \min \left\{ 1, \frac{D}{\|\Delta_t - \eta \tilde{g}_t\|} \right\} \cdot \left( \Delta_t - \eta \tilde{g}_t \right)$$

is precisely the online project gradient descent update rule, with respect to linear losses of the form $\ell_t(\cdot) = \langle \tilde{g}_t, \cdot \rangle$, over the ball of radius $D$ around the origin. Accordingly, recalling that $\mathbb{E} \| \tilde{g}_t - \nabla h(z_t) \|^2 \leq G_1^2$, combining the linearity of expectation with the standard regret analysis of online linear optimization (cf. Hazan, 2016) gives the following:

**Lemma B.1.** *By setting $\eta = \frac{D}{G_1 \sqrt{M}}$, for any $u \in \mathbb{R}^d$ with $\|u\| \leq D$ it holds that*

$$\mathbb{E}_{\tilde{g}_1, \ldots, \tilde{g}_M} \left[ \sum_{m=1}^{M} \langle \tilde{g}_m, \Delta_m - u \rangle \right] \leq \tfrac{3}{2} D G_1 \sqrt{M}.$$

Back to analyzing Algorithm 2, since $x_t = x_{t-1} + \Delta_t$ it holds that

$$
\begin{aligned}
h(x_t) - h(x_{t-1}) &= \int_0^1 \langle \nabla h(x_{t-1} + s\Delta_t), \Delta_t \rangle \, ds \\
&= \mathbb{E}_{s_t \sim \text{Unif}[0,1]} \left[ \langle \nabla h(x_{t-1} + s_t \Delta_t), \Delta_t \rangle \right] = \mathbb{E}_{s_t} \left[ \langle \nabla h(z_t), \Delta_t \rangle \right].
\end{aligned}
$$

Note that $\langle \nabla h(z_t), \Delta_t \rangle = \langle \nabla h(z_t), u \rangle + \langle \tilde{g}_t, \Delta_t - u \rangle + \langle \nabla h(z_t) - \tilde{g}_t, \Delta_t - u \rangle$, so by summing over $t \in [T] = [K \times M]$, we get for any fixed sequence $u_1, \ldots, u_K \in \mathbb{R}^d$ :

$$
\begin{aligned}
\inf h \leq h(x_T) &\leq h(x_0) + \sum_{t=1}^{T} \mathbb{E} \left[ \langle \nabla h(z_t), \Delta_t \rangle \right] \\
&= h(x_0) + \sum_{k=1}^{K} \sum_{m=1}^{M} \mathbb{E} \left[ \langle \tilde{g}_{(k-1)M+m}, \Delta_{(k-1)M+m} - u_k \rangle \right] \\
&\quad + \sum_{k=1}^{K} \sum_{m=1}^{M} \mathbb{E} \left[ \langle \nabla h(z_{(k-1)M+m}), u_k \rangle \right] + \sum_{t=1}^{T} \mathbb{E}[\langle \nabla h(z_t) - \tilde{g}_t, \Delta_t - u \rangle] \\
&\leq h(x_0) + \tfrac{3}{2} K D G_1 \sqrt{M} + \sum_{k=1}^{K} \sum_{m=1}^{M} \mathbb{E} \left[ \langle \nabla h(z_{(k-1)M+m}), u_k \rangle \right] + G_0 DT,
\end{aligned}
$$

where the last inequality follows from applying Lemma B.1 to each $M$ consecutive iterates, and combining the bias bound $\mathbb{E} \| \tilde{g}_t - \nabla h(z_t) \| \leq G_0$ with Cauchy-Schwarz.

Letting $u_k := -D \frac{\sum_{m=1}^{M} \nabla h(z_{(k-1)M+m})}{\left\| \sum_{m=1}^{M} \nabla h(z_{(k-1)M+m}) \right\|}$, rearranging and dividing by $DT = DKM$, we obtain

$$\frac{1}{K} \sum_{k=1}^{K} \mathbb{E} \left\| \frac{1}{M} \sum_{m=1}^{M} \nabla h(z_{(k-1)M+m}) \right\| \leq \frac{h(x_0) - \inf h}{DT} + \frac{3G_1}{2\sqrt{M}} + G_0. \tag{16}$$

Finally, note that for all $k \in [K], m \in [M] : \left\| z_{(k-1)M+m} - \overline{x}_k \right\| \leq MD \leq \alpha$ since the clipping operation ensures each iterate is at most of distance $D$ to its predecessor, and therefore $\nabla h(z_{(k-1)M+m}) \in \partial_\alpha h(\overline{x}_k)$. Since the set $\partial_\alpha h(\cdot)$ is convex by definition, we further see that

$$\frac{1}{M} \sum_{m=1}^{M} \nabla h(z_{(k-1)M+m}) \in \partial_\alpha h(\overline{x}_k) ,$$

and hence by Eq. (16) we get

$$\mathbb{E} \left\| \overline{\partial}_\alpha h(x^{\text{out}}) \right\| = \frac{1}{K} \sum_{k=1}^{K} \mathbb{E} \left\| \overline{\partial}_\alpha h(\overline{x}_k) \right\| \leq \frac{h(x_0) - \inf h}{DT} + \frac{3G_1}{2\sqrt{M}} + G_0.$$

## C    FIRST-ORDER ALGORITHM

In this appendix, our goal is to show that the zero-order algorithms presented in the main text can be replaced by first-order algorithms with the sample complexity, and reduced oracle complexity.

The simple idea is to replace the zero-order gradient estimator from Eq. (4) by the first-order estimator

$$\tilde{\nabla} f_\alpha(x; \xi) = \frac{1}{m} \sum_{j=1}^m \nabla f(x + \alpha y_j; \xi), \ \ y_1, \dots, y_m \stackrel{iid}{\sim} \mathrm{Unif}(\mathbb{S}^{d-1}). \tag{17}$$

While this estimator has the same expectation as the zero-order variant, the key difference lies in the fact that its subGaussian norm is substantially smaller (as it does not depend on $d$), hence smaller $m$ suffices for concentration. This observation enables reducing the oracle complexity, while ensuring the same sample complexity guarantee as in the main text.

We fully analyze here a single-pass first-order oracle, presented in Algorithm 5, which can be used in Algorithm 2, similarly to Section 3. We note that a similar analysis can be applied to the multi-pass oracle of Section 4, once again by replacing Eq. (4) by Eq. (17).

As in Section 3, we will present the basic properties of this oracle. We will then plug these into Algorithm 2, leading to the main result of this section, Theorem C.4.

---

**Algorithm 5** First-order instantiation of $\mathcal{O}(z_t)$ in Line 7 of Algorithm 2

1: **Input:** Current iterate $z_t$, time $t \in \mathbb{N}$, period length $\Sigma \in \mathbb{N}$, accuracy parameter $\alpha > 0$, batch sizes $B_1, B_2 \in \mathbb{N}$, gradient validation size $m \in \mathbb{N}$, noise level $\sigma > 0$.
2: **if** $t \bmod \Sigma = 1$ **then**
3:     Sample minibatch $S_t$ of size $B_1$ among unused samples
4:     Sample $y_1, \dots, y_{B_1} \stackrel{iid}{\sim} \mathrm{Unif}(\mathbb{B}_\alpha)$
5:     $g_t = \frac{1}{B_1} \sum_{\xi_i \in S_t} \nabla f(z_t + y_i; \xi_i)$
6: **else**
7:     Sample minibatch $S_t$ of size $B_2$ among unused samples
8:     **for** each sample $\xi_i \in S_t$ **do**
9:         Sample $y_1, \dots, y_{2m} \stackrel{iid}{\sim} \mathrm{Unif}(\mathbb{B}_\alpha)$
10:        $\tilde{\nabla} f(z_t; \xi_i) = \frac{1}{m} \sum_{j=1}^m \nabla f(z_t + y_j; \xi_i)$
11:        $\tilde{\nabla} f(z_{t-1}; \xi_i) = \frac{1}{m} \sum_{j=m+1}^{2m} \nabla f(z_{t-1} + y_j; \xi_i)$
12:    **end for**
13:    $g_t = g_{t-1} + \frac{1}{B_2} \sum_{\xi_i \in S_t} (\tilde{\nabla} f(z_t; \xi_i) - \tilde{\nabla} f(z_{t-1}; \xi_i))$
14: **end if**
15: **Return** $\tilde{g}_t = g_t + \mathrm{TREE}(\sigma, \Sigma)(t \bmod \Sigma)$

---

**Lemma C.1** (Sensitivity). *Consider the gradient oracle $\mathcal{O}(\cdot)$ in Algorithm 5 when acting on two neighboring minibatches $S_t$ and $S_t'$, and correspondingly producing $g_t$ and $g_t'$, respectively. If $t \bmod \Sigma = 1$, then*

$$\|g_t - g_t'\| \leq \frac{L}{B_1}.$$

*Otherwise, conditioned on $g_{t-1} = g_{t-1}'$, we have with probability at least $1 - \delta/2$ :*

$$\|g_t - g_t'\| \lesssim \frac{L\sqrt{d}D}{\alpha B_2} + \frac{L\sqrt{\log(dB_2/\delta)}}{\sqrt{m}}.$$

With the sensitivity bound given by Lemma C.1, we easily derive the privacy guarantee of our algorithm from the Tree Mechanism (Proposition 2.5).

**Lemma C.2** (Privacy). *Running Algorithm 5 with* $m = O(\log(dB_2/\delta)\frac{B_2^2\alpha^2}{D^2d})$ *and* $\sigma = O(\frac{L\sqrt{\log(1/\delta)}}{B_1\varepsilon} + \frac{LD\sqrt{d\log(1/\delta)}}{\alpha B_2\varepsilon})$ *is* $(\varepsilon, \delta)$-*DP.*

*Proof.* By Lemma C.1 and our assignment of $m$, we know that with probability at least $1 - \delta/2$, for any $t$, we have

$$\|g_t - g_t'\| \lesssim \frac{L}{B_1} + \frac{L\sqrt{d}D}{\alpha B_2}.$$

Then the privacy guarantee follows from the Tree Mechanism (Proposition 2.5).

$\square$

We next provide the required variance bound on the gradient oracle.

**Lemma C.3** (Variance). *In Algorithm 5, for all $t$ it holds that*

$$\mathbb{E}\|\tilde{g}_t - \nabla F_\alpha(z_t)\|^2 \lesssim \frac{L^2}{B_1} + \frac{L^2 dD^2 \Sigma}{\alpha^2 B_2} + \sigma^2 d \log \Sigma + \frac{L^2 \Sigma}{mB_2},$$

$$\mathbb{E}\|\tilde{g}_t\|^2 \lesssim L^2 + \frac{L^2 dD^2 \Sigma}{\alpha^2 B_2} + \sigma^2 d \log \Sigma + \frac{L^2 \Sigma}{mB_2}.$$

Having set up the required bounds, we can prove our main result for the first-order setting.

**Theorem C.4** (First-order). *Suppose $F(x_0) - \inf_x F(x) \leq \Phi$, that Assumption 2.2 holds, and let $\alpha, \beta, \delta, \varepsilon > 0$ such that $\alpha \leq \frac{\Phi}{L}$. Then setting $B_1 = \Sigma$, $B_2 = 1$, $M = \alpha/4D$, $m = \widetilde{O}(\frac{B_2^2 \alpha^2}{D^2 d})$, $\sigma = \widetilde{O}(\frac{L}{B_1 \varepsilon} + \frac{LD\sqrt{d}}{\alpha B_2})$, $\Sigma = \widetilde{\Theta}((\frac{\alpha}{\varepsilon D})^{2/3})$, $D = \widetilde{\Theta}(\min\{(\frac{\Phi^2 \alpha}{L^2 T^2})^{1/3}, (\frac{\Phi \alpha \varepsilon}{dLT})^{1/2}, (\frac{\Phi^3 \alpha^2 \varepsilon}{d^{3/2} L^3 T^3})^{1/5}\})$, $T = \Theta(n)$, and running Algorithm 2 with Algorithm 5 as the oracle subroutine, is $(\varepsilon, \delta)$-DP. Furthermore, its output satisfies $\mathbb{E}\|\bar{\partial}_{2\alpha} F(x^{out})\| \leq \beta$ as long as*

$$n = \widetilde{\Omega}\left(\frac{\Phi L^2}{\alpha \beta^3} + \frac{\Phi Ld}{\varepsilon \alpha \beta^2} + \frac{\Phi L^{3/2} d^{3/4}}{\varepsilon^{1/2} \alpha \beta^{5/2}}\right).$$

**Remark C.5** (Oracle complexity). Compared to the zero-order result given by Theorem 3.1, we see that the number of calls to $\mathcal{O}(\,\cdot\,)$, namely $T$, is on the same order, and that in both cases the amortized oracle complexity of $\mathcal{O}(\,\cdot\,)$ is $O(m)$. The difference between the settings is that the first-order oracle instantiation sets $m$ to be $\widetilde{\Omega}(d^2)$ times smaller than its zero-order counterpart, and hence we gain this multiplicative factor in the overall oracle complexity.

*Proof of Theorem C.4.* The privacy guarantee follows directly from Lemma C.2, by noting that our parameter assignment implies $B_1 T/\Sigma + B_2 T = O(n)$, hence it allows letting $T = \Theta(n)$ while never re-using samples.

As to the sample complexity, note that our parameter assignment ensures that

$$G_1 = O(G_0 + L),$$

$$G_0 = \widetilde{O}\left(\frac{LDd^{1/2}\Sigma^{1/2}}{\alpha} + \frac{Ld^{1/2}}{\Sigma \varepsilon} + \frac{LDd}{\alpha \varepsilon}\right),$$

similarly to Eq. (7) and Eq. (9) in the proof of Theorem 3.1. The rest of the proof is therefore exactly the same as for Theorem 3.1. $\square$

## C.1 PROOFS FROM APPENDIX C

*Proof of Lemma C.1.* The case when $t \mod \Sigma = 1$ trivially follows the Lipschitz assumption. Thus we will consider the more involved case. For any $\xi \in S_t$, by a standard sub-Gaussian bound (Theorem E.2) we have

$$\Pr\left[\|\tilde{\nabla} f(z_t; \xi) - \nabla f_\alpha(z_t; \xi)\| \leq \frac{L\sqrt{\log(8dB_2/\delta)}}{\sqrt{m}}\right] \geq 1 - \delta/8B_2,$$

so by the union bound, we get that with probability at least $1 - \delta/8$, for all $\xi_i \in S_t$ :

$$\|\tilde{\nabla} f(z_t; \xi) - \nabla f_\alpha(z_t; \xi)\| \leq \frac{L\sqrt{\log(8dB_2/\delta)}}{\sqrt{m}}. \tag{18}$$

Hence,

$$\|g_t - g'_t\| \leq \left\| \frac{1}{B_2} \sum_{\xi \in S_t} \left( (\tilde{\nabla} f(z_t; \xi) - \tilde{\nabla} f(z_{t-1}; \xi_i)) - (\nabla f_\alpha(z_t; \xi)) - \nabla f_\alpha(z_{t-1}; \xi)) \right) \right\|$$

$$+ \left\| \frac{1}{B_2} \sum_{\xi \in S_t} \left( (\nabla f_\alpha(z_t; \xi) - \nabla f_\alpha(z_{t-1}; \xi)) - \sum_{\xi' \in S'_t} (\nabla f_\alpha(z_t; \xi') - \nabla f_\alpha(z_t; \xi')) \right) \right\|$$

$$+ \left\| \frac{1}{B_2} \sum_{\xi' \in S'_t} \left( (\tilde{\nabla} f(z_t; \xi') - \tilde{\nabla} f(z_{t-1}; \xi')) - (\nabla f_\alpha(z_t; \xi') - \nabla f_\alpha(z_{t-1}; \xi')) \right) \right\|$$

$$\lesssim \frac{L\sqrt{d}D}{\alpha B_2} + \frac{L\sqrt{\log(dB_2/\delta)}}{\sqrt{m}},$$

where the last inequality step is due to the smoothness of $f_\alpha$ (Fact 2.3) combined with the fact that $\|z_t - z_{t-1}\| \leq 2D$, and Eq. (18).

$\square$

*Proof of Lemma C.3.* Applying by Proposition 2.5, we have

$$\mathbb{E}\|\tilde{g}_t - \nabla F_\alpha(z_t)\|^2 \lesssim \mathbb{E}\|\tilde{g}_t - g_t\|^2 + \mathbb{E}\|g_t - \nabla F_\alpha(z_t)\|^2 \lesssim d\sigma^2 \log \Sigma + \mathbb{E}\|g_t - \nabla F_\alpha(z_t)\|^2,$$

and also since $\mathbb{E}[g_t] = \nabla F_\alpha(z_t)$ and $\|\nabla F_\alpha(z_t)\| \leq L$, we have

$$\mathbb{E}\|\tilde{g}_t\|^2 \lesssim \mathbb{E}\|g_t\|^2 + d\sigma^2 \log \Sigma \lesssim \mathbb{E}\|g_t - \nabla F_\alpha(z_t)\|^2 + L^2 + d\sigma^2 \log \Sigma.$$

We therefore see that both claimed bounds will follow from bounding $\mathbb{E}\|g_t - \nabla F_\alpha(z_t)\|^2$.

To that end, denote by $t_0 \leq t$ the largest integer such that $t_0 \bmod \Sigma = 1$, and note that $t - t_0 < \Sigma$. Further denote $\Delta_j := g_j - g_{j-1}$. Then we have

$$\mathbb{E}\|g_t - \nabla F_\alpha(z_t)\|^2 = \mathbb{E}\left\| g_{t_0} + \sum_{j=t_0+1}^{t} \Delta_j - \left( \sum_{j=t_0+1}^{t} (\nabla F_\alpha(z_j) - \nabla F_\alpha(z_{j-1})) + \nabla F_\alpha(z_{t_0}) \right) \right\|^2$$

$$= \mathbb{E}\|g_{t_0} - \nabla F_\alpha(z_{t_0})\|^2 + \sum_{j=t_0}^{t} \mathbb{E}\|\Delta_j - (\nabla F_\alpha(z_j) - \nabla F_\alpha(z_{j-1}))\|^2,$$

$$\lesssim \frac{L^2}{B_1} + \sum_{j=t_0}^{t} \underbrace{\mathbb{E}\|\Delta_j - (\nabla F_\alpha(z_j) - \nabla F_\alpha(z_{j-1}))\|^2}_{(\star)} \tag{19}$$

where the second equality is due to the cross terms having zero mean. Moreover, we have

$$(\star) = \mathbb{E}\left\| \frac{1}{B_2} \sum_{\xi \in S_t} (\tilde{\nabla} f(z_j; \xi) - \tilde{\nabla} f(z_{j-1}; \xi)) - (\nabla F_\alpha(z_j) - \nabla F_\alpha(z_{j-1})) \right\|^2$$

$$= \frac{1}{B_2^2} \sum_{\xi \in S_t} \mathbb{E}\|(\tilde{\nabla} f(z_j; \xi) - \tilde{\nabla} f(z_{j-1}; \xi)) - (\nabla F_\alpha(z_j) - \nabla F_\alpha(z_{j-1}))\|^2$$

$$\lesssim \frac{1}{B_2^2} \sum_{\xi \in S_t} \left( \mathbb{E}\|\tilde{\nabla} f(z_j; \xi) - \nabla f_\alpha(z_j; \xi)\|^2 + \mathbb{E}\|\tilde{\nabla} f(z_{j-1}; \xi) - \nabla f_\alpha(z_{j-1}; \xi)\|^2 \right.$$

$$\left. + \mathbb{E}\|(\nabla f_\alpha(z_j; \xi) - \nabla f_\alpha(z_{j-1}; \xi)) - (\nabla F_\alpha(z_j) - \nabla F_\alpha(z_{j-1}))\| \right)^2$$

$$\lesssim \frac{L^2}{mB_2} + \frac{dL^2D^2}{\alpha^2 B_2},$$

which plugged into Eq. (19) completes the proof by recalling that $t - t_0 \leq \Sigma$.

$\square$

# D  EVEN BETTER SAMPLE COMPLEXITY VIA OPTIMAL SMOOTHING

In this Appendix, our aim is to provide evidence that the sample complexities of NSNC DP optimization obtained in our work are likely improvable, at least with a computationally inefficient method. This approach is inspired by Lowy et al. (2024), which in the context of smooth optimization, showed significant sample complexity gains using algorithms with exponential runtime. As as we will show, a similar phenomenon might hold for nonsmooth optimization. To that end, we propose a slight relaxation of Goldstein-stationarity, and show it can be achieved using less samples via an exponential time algorithm.

## D.1  RELAXATION OF GOLDSTEIN-STATIONARITY

Recall that $x \in \mathbb{R}^d$ is called an $(\alpha, \beta)$-Goldstein stationary point of an objective $F(x) = \mathbb{E}_\xi[f(x; \xi)]$ if there exist $y_1, \ldots, y_k \in \mathbb{B}(x, \alpha)$ and convex coefficients $(\lambda_i)_{i=1}^k$ so that $\|\sum_{i \in [k]} \lambda_i \mathbb{E}_\xi[\nabla f(y_i; \xi)]\| \leq \beta$. Arguably, the two most important properties satisfied by this definition are that

(i) If $f(x; \xi)$ are $L$-smooth, any $(\alpha, \beta)$-stationary point is $O(\alpha + \beta)$-stationary.

(ii) If $\|\overline{\partial}_\alpha F(x)\| \neq 0$, then $F\left(x - \frac{\alpha}{\|\overline{\partial}_\alpha F(x)\|} \overline{\partial}_\alpha F(x)\right) \leq F(x) - \alpha \|\overline{\partial}_\alpha F(x)\|$.

The first property shows that Goldstein-stationarity reduces to ("classic") stationarity under smoothness. The second, known as Goldstein's descent lemma (Goldstein, 1977), is a generalization of the classic descent lemma for smooth functions.

It is easy to see that Goldstein-stationarity is equivalent to the existence of a distribution $P$ supported over $\mathbb{B}(x, \alpha)$, such that $\|\mathbb{E}_{\xi, y \sim P}[\nabla f(y; \xi)]\| \leq \beta$. We will now define a relaxation of Goldstein-stationarity that is easily verified to satisfy both of the aforementioned properties.

**Definition D.1.** We call a point $x \in \mathbb{R}^d$ an $(\alpha, \beta)$-*component-wise* Goldstein-stationary point of $F(x) = \mathbb{E}_\xi[f(x; \xi)]$ if there exist distributions $P_\xi$ supported over $\mathbb{B}(x, \alpha)$, such that $\|\mathbb{E}_{\xi, y \sim P_\xi}[\nabla f(y; \xi)]\| \leq \beta$.

In other words, the definition above allows the sampled points $y_1, \ldots, y_k$ in the vicinity of $x$ to vary for different components, and as before, the sampled gradient must have small expected norm. We next show that this relaxed stationarity notion allows improving the sample complexity of DP NSNC optimization.

## D.2  OPTIMAL SMOOTHING AND FASTER ALGORITHM

In the previous sections, given an objective $f$, we used the fact that Goldstein-stationary points of the randomized smoothing $f_\alpha$ correspond to Goldstein-stationary point of $f$, and therefore constructed private gradient oracles of $f_\alpha$, which is $O(\sqrt{d}/\alpha)$-smooth. Consequently, the sensitivity of the gradient oracle had a $\sqrt{d}$ dimension dependence (as seen in Lemma 3.3), thus affecting the overall sample complexity.

Instead of randomized smoothing, we now consider the Lasry-Lions (LL) smoothing (Lasry & Lions, 1986), a method that smooths Lipschitz functions in a dimension independent manner, which we now recall. Given $h : \mathbb{R}^d \to \mathbb{R}$, denote the so-called Moreau envelope

$$M_\lambda(h)(x) := \min_y \left[ h(y) + \frac{1}{2\lambda} \|y - x\|^2 \right],$$

and the Lasry-Lions smoothing:

$$\tilde{h}_{\lambda\text{LL}}(x) := -M_\lambda(-M_{2\lambda}(h))(x) = \max_z \min_y \left[ h(z) + \frac{1}{4\lambda} \|z - y\|^2 - \frac{1}{2\lambda} \|y - x\|^2 \right]. \quad (20)$$

**Fact D.2.** *[Lasry & Lions, 1986; Attouch & Aze, 1993] Suppose $h : \mathbb{R}^d \to \mathbb{R}$ is $L$-Lipschitz. Then: (i) $\tilde{h}_{\lambda\text{LL}}$ is $L$-Lipschitz; (ii) $|\tilde{h}_{\lambda\text{LL}}(x) - h(x)| \leq L\lambda$ for any $x \in \mathbb{R}^d$; (iii) $\arg\min \tilde{h}_{\lambda\text{LL}} = \arg\min h$; (iv) $\tilde{h}_{\lambda\text{LL}}$ is $O(L/\lambda)$-smooth.*

The key difference between LL-smoothing and randomized smoothing is that the smoothness constant of LL-smoothing is dimension independent. By solving the optimization problem in Eq. Eq. (20), it is clear that the values, and therefore gradients, of $\tilde{f}_{\lambda\text{LL}}(x; \xi_i)$ can be obtained up to arbitrarily high accuracy. Notably, it was shown by Kornowski & Shamir (2022) that solving this problem requires, in general, an exponential number of oracle calls to the original function.

Nonetheless, computational considerations aside, it is not even clear that the LL smoothing helps in terms of finding Goldstein-stationary points of the original function, which was previously shown for randomized smoothing (Lemma 2.4). This is the purpose of the following result, which we prove:

**Lemma D.3.** *If $h$ is $L$-Lipschitz, then any $\beta$-stationary point of $\tilde{h}_{\lambda\text{LL}}$ is a $(3\lambda L, \beta)$-Goldstein stationary point of $h$.*

Given the lemma above, we are able to utilize smooth algorithms for finding stationary points, and convert the guarantee to Goldstein-stationary points of our objective of interest. Specifically, we will invoke the following result.

**Proposition D.4** (Lowy et al., 2024). *Given an ERM objective $\tilde{F}(x) = \frac{1}{n}\sum_{i=1}^{n}\tilde{f}(x; \xi_i)$ with $L_0$-Lipschitz and $L_1$-smooth components, and an initial point $x_0 \in \mathbb{R}^d$ such that $\text{dist}(x_0, \arg\min\tilde{F}) \leq R$, there's an $(\varepsilon, \delta)$-DP algorithm that returns $\tilde{x}^{\text{out}}$ with $\mathbb{E}\|\nabla\tilde{F}(\tilde{x}^{\text{out}})\| = \widetilde{O}\left(\frac{R^{1/3}L_0^{2/3}L_1^{1/3}d^{2/3}}{n\varepsilon} + \frac{L_0\sqrt{d}}{n\varepsilon}\right).$*

We remark that we assume for simplicity that $\text{dist}(x_0, \arg\min\widehat{F}^{\mathcal{D}}) = \text{dist}(x_0, \arg\min\tilde{F}) \leq R$, though the analysis extends to that case where $R$ is the initial distance to a point with sufficiently small loss (e.g., if the infimum is not attained). Overall, by setting $\lambda = \alpha/3L$, and combining Fact D.2, Lemma D.3 and Proposition D.4, we get the following:

**Theorem D.5.** *Under Assumption 2.2, suppose $\text{dist}(x_0, \arg\min\widehat{F}^{\mathcal{D}}) \leq R$. Then there is an $(\varepsilon, \delta)$-DP algorithm that outputs $x^{\text{out}}$ satisfying $(\alpha, \beta)$-component-wise Goldstein-stationarity (in expectation) as long as*

$$n = \widetilde{\Omega}\left(\frac{R^{1/3}L^{4/3}d^{2/3}}{\varepsilon\alpha^{1/3}\beta}\right).$$

### D.3 PROOFS FROM APPENDIX D

*Proof of Lemma D.3.* Suppose $x$ is a $\beta$-stationary point of $\tilde{h}_{\lambda\text{LL}}$. Let $z^* \in \mathbb{R}^d$ be the solution of the maximization problem defining the LL smoothing. By (Attouch & Aze, 1993, Remark 4.3.e), $z^*$ is uniquely defined, and satisfies

$$\nabla\tilde{h}_{\lambda\text{LL}}(x) \in \partial(M_{2\lambda}(h))(z^*). \tag{21}$$

Further denote $\mathcal{Y}^* := \arg\min_y\left[h(y) + \frac{1}{4\lambda}\|z^* - y\|^2\right] \subseteq \mathbb{R}^d$. Rearranging the definition of the Moreau envelope by expanding the square, we see that

$$M_{2\lambda}(h)(z^*) = \frac{1}{4\lambda}\|z^*\|^2 - \frac{1}{2\lambda}\max_y\left[\langle z^*, y\rangle - 2\lambda h(y) - \frac{1}{2}\|y\|^2\right],$$

from which we get

$$\partial M_{2\lambda}h(z^*) = \frac{1}{2\lambda}z^* - \frac{1}{2\lambda}\text{conv}\{y^* : y^* \in \mathcal{Y}^*\} = \text{conv}\left\{\frac{1}{2\lambda}(z^* - y^*) : y^* \in \mathcal{Y}^*\right\}. \tag{22}$$

Furthermore, for any $y^* \in \mathcal{Y}^*$, by first-order optimality it holds that

$$0 \in \partial\left[h(y^*) + \frac{1}{4\lambda}\|y^* - z^*\|^2\right] \subseteq \partial h(y^*) + \frac{1}{2\lambda}(y^* - z^*),$$

and therefore

$$\frac{1}{2\lambda}(z^* - y^*) \in \partial h(y^*). \tag{23}$$

By combining Eq. (21), Eq. (22) and Eq. (23) we conclude that

$$\nabla \tilde{h}_{\lambda\text{LL}}(x) \in \partial M_{2\lambda}h(z^*) \subseteq \text{conv}\left\{\partial h(y^*) : y^* \in \mathcal{Y}^*\right\} \subseteq \partial_r h(x),$$

where the last holds for $r := \max_{y^* \in \mathcal{Y}^*} \|x - y^*\|$. Therefore, recalling that $\|\nabla \tilde{h}_{\lambda\text{LL}}(x)\| \leq \beta$, all that remains is to bound $r$.

To that end, it clearly holds that $r \leq \|x - z^*\| + \max_{y^* \in \mathcal{Y}^*} \|z^* - y^*\|$. Furthermore, by (Attouch & Aze, 1993, Remark 4.3.e) it holds that $z^* - x = \lambda \nabla \tilde{h}_{\lambda\text{LL}}(x)$ which implies $\|x - z^*\| = \lambda\beta$. As to the second summand, by Eq. (22) it holds that $\max_{y^* \in \mathcal{Y}^*} \|z^* - y^*\| \leq 2\lambda \cdot \max_{g \in \partial M_{2\lambda}h(z^*)} \|g\| \leq 2\lambda L$, by the fact that $M_{2\lambda}(h)$ is $L$-Lipschitz. Overall $r \leq \lambda\beta + 2\lambda L$, and as we can assume without loss of generality that $\beta \leq L$ since otherwise the claim is trivially true (note that all points are $L$ stationary), this completes the proof.

□

## E  CONCENTRATION LEMMA FOR VECTORS WITH SUB-GAUSSIAN NORM

Here we recall a standard concentration bound for vectors with sub-Gaussian norm, which notably applies in particular to bounded random vectors.

**Definition E.1** (Norm-sub-Gaussian). We say a random vector $X \in \mathbb{R}^d$ is $\zeta$-norm-sub-Gaussian for $\zeta > 0$, if $\Pr[\|X - \mathbb{E}X\| \geq t] \leq 2e^{-t^2/2\zeta^2}$ for all $t \geq 0$.

**Theorem E.2** (Hoeffding-type inequality for norm-subGaussian, Jin et al., 2019). *Let $X_1, \cdots, X_k \in \mathbb{R}^d$ be random vectors, and let $\mathcal{F}_i = \sigma(X_1, \cdots, X_i)$ for $i \in [k]$ be the corresponding filtration. Suppose for each $i \in [k]$, $X_i \mid \mathcal{F}_{i-1}$ is zero-mean $\zeta_i$-norm-sub-Gaussian. Then, there exists an absolute constant $c > 0$, such that for any $\gamma > 0$:*

$$\Pr\left[\left\|\sum_{i \in [k]} X_i\right\| \geq c\sqrt{\log(d/\gamma) \sum_{i \in [k]} \zeta_i^2}\right] \leq \gamma.$$

