# OpenReview forum: "Improved Sample Complexity for Private Nonsmooth Nonconvex Optimization"
_ICLR.cc/2025/Conference — Submitted to ICLR 2025_

### Official Review · Reviewer_WTuS · 2024-11-02

**Soundness:** 3
**Presentation:** 3
**Contribution:** 2
**Rating:** 5
**Confidence:** 4

**Summary:**

This paper addresses differential privacy (DP) in nonsmooth, nonconvex optimization, aiming to improve sample complexity for finding Goldstein-stationary points in such challenging settings. Traditional DP optimization methods often assume convexity or smoothness, but this work proposes new algorithms that can handle nonsmooth nonconvex (NSNC) objectives.
### Key Contributions

1. **Single-Pass Algorithm**
   The authors present a single-pass (ε, δ)-DP algorithm that finds an (α, β)-stationary point with improved sample complexity. This algorithm reduces dimensional dependence by a factor of \(\Omega(\sqrt{d})\) over previous approaches, making DP optimization feasible in high-dimensional settings while maintaining privacy guarantees.

2. **Multi-Pass Algorithm**
   A multi-pass ERM-based algorithm further enhances sample efficiency, allowing the algorithm to iterate over the data multiple times and achieve sublinear dimension-dependent sample complexity. This approach improves convergence while satisfying DP constraints.

3. **Generalization from ERM to Population Loss**
   The authors establish that Goldstein-stationarity achieved on empirical loss also applies to the population loss with high probability. This result expands the utility of their approach by ensuring that empirical results generalize to the population.


The proposed algorithms make notable progress in DP optimization for NSNC problems, improving sample efficiency while maintaining privacy. This advancement is valuable for practical applications where data privacy is essential, especially in high-dimensional machine learning settings. Additionally, the generalization result strengthens the applicability of Goldstein-stationary points beyond empirical settings.

**Strengths:**

1. **Significant Improvement in Sample Complexity**
   The paper offers a substantial reduction in sample complexity for differentially private (DP) nonsmooth nonconvex (NSNC) optimization. The single-pass algorithm achieves a lower dependence on dimension \(d\) compared to prior work, which is highly impactful for high-dimensional problems in machine learning.

2. **Innovative Use of Goldstein-Stationarity**
   By focusing on Goldstein-stationary points, the authors leverage a nuanced stationarity condition suitable for nonsmooth nonconvex optimization, allowing for more practical solutions where traditional gradient-based methods fall short. This approach builds on and expands the utility of Goldstein-stationarity in DP settings.

3. **Generalization from Empirical to Population Loss**
   The paper addresses a theoretical gap by proving that empirical guarantees of Goldstein-stationarity translate to the population loss. This generalization strengthens the theoretical foundation and practical relevance of the proposed algorithms, as it ensures that results on empirical data apply to broader distributions.

4. **Applicability to Real-World DP Machine Learning Tasks**
   The proposed algorithms are zero-order (using only function evaluations) and thus avoid the need for gradient information, making them suitable for a wider range of machine learning models that may have nonsmooth, nonconvex loss landscapes. This approach is particularly beneficial in privacy-sensitive applications like federated learning.

5. **Novel Dimension-Independent Term**
   The single-pass algorithm introduces a dimension-independent term in the "non-private" component of the sample complexity, challenging previous assumptions in DP optimization for NSNC objectives. This innovation indicates potential for further sample complexity improvements and opens new directions for DP research in nonconvex settings.

**Weaknesses:**

1. There is some typo, for instance, line 077 the last word should be perform.

2. Randomized Smoothing is an ordinary technique used in this setting, and I wonder the novelty except for this to deal with the non-smooth setting.

**Questions:**

1. Is the result in the paper tight? In other words, is there a lower bound provided?

2. What is the key challenge in improving the result by at least $\(\sqrt{d}\)$? Specifically, how does this improvement compare to the results in the referenced work?

3. What role does the (α, β)-Goldstein stationary point play in this paper?

4. What is the novelty of this paper compared to previous works?

5. Can you explain more about the result regarding the non-private term and private and how they contribute to the final result?

---

> ### Author Response · Authors · 2024-11-20
> **Response to reviewer WTuS**
>
> We thank the reviewer for their time, for appreciating the strengths of our work, and appreciate their constructive efforts in helping us strengthen it.
>
> We address the raised questions:
>
> 1. It is currently not known whether the result of this paper is tight, due to lacking lower bounds in the private regime. The non-private term is indeed optimal (as indicated by the lower bound by Cutkosky et al.), whereas the exact private terms are not even known in the well-studied smooth nonconvex case.
>
> 2. As we discuss in the paper, we identify several challenges in order to improve previous results. The first of which is to decrease the sensitivity of the gradients estimator which allows adding less noise in order to privatize. Moreover, we then also further improve the results by utilizing the empirical objective for which we derive tighter guarantees, and prove that these also generalize to the population, which no previous work has done in this context. We further discuss these in the top common comment.
>
> 3. Goldstein-stationarity has recently emerged as a popular framework for designing and analyzing optimization algorithms for nonsmooth nonconvex objectives, which is an important regime in modern deep learning applications. The reason for this is that function-value minimization as well as stationarity are impossible to efficiently achieve in the non-smooth non-convex setting, and Goldstein stationarity is the strongest known optimality condition that can be efficiently achieved. While many works studied this notion without privacy, only one previous paper studied this notion in the private setting. In our paper, we substantially improve upon these previous results, and along the way develop techniques (e.g., generalization of Goldstein-stationary points) that may find uses beyond private optimization.
>
> 4. As pointed out throughout the paper (and also acknowledged by the reviewer, e.g. strengths #3 and #5), there are several novel aspects in our analysis that allow us to derive significantly better results than previous works on this subject. These include reducing the sensitivity of the gradient estimator, and also generalizing from the empirical loss to the population loss. Please note the top common comment for further discussion.
>
> 5. In private stochastic optimization, the sample complexity consists of terms that depend on the privacy parameters, and terms that are independent of them (referred to as “non-private” terms). We significantly improve both terms in our work, and emphasize that the non-private term we get is even better than previously thought to be possible (and in particular, optimal).
>
> Given that the reviewer acknowledges the significant contributions of our work, we kindly ask them to consider reevaluating their rating in light of these clarifications. We hope we have been able to address the raised questions, and please let us know in case of any additional questions or feedback.

---

> > ### Author Response · Authors · 2024-11-26
> >
> > Dear Reviewer,
> >
> > Thank you once again for your detailed comments. We would greatly appreciate knowing if our responses have adequately addressed your concerns and questions, and we would also be happy to engage in any further discussions if needed.

---

> > ### Comment · Reviewer_WTuS · 2024-11-26
> >
> > Thanks for your reply, I will keep my score.

---

### Official Review · Reviewer_hF98 · 2024-11-03

**Soundness:** 2
**Presentation:** 2
**Contribution:** 2
**Rating:** 5
**Confidence:** 4

**Summary:**

This paper explores differentially private (DP) optimization algorithms for stochastic and empirical objectives that are non-smooth and non-convex, presenting methods that achieve Goldstein-stationary points with improved sample complexity bounds compared to prior work. The authors introduce a single-pass ($\epsilon$,$\delta$)-DP algorithm capable of producing ($\alpha$,$\beta$)-stationary points. Subsequently, they propose a multi-pass, polynomial-time algorithm that further refines sample efficiency by designing an effective ERM algorithm and demonstrating that Goldstein-stationary points can generalize from the empirical to the population loss.

**Strengths:**

The paper studies an important problem in DP non-convex optimization, and achieves improved sample complexity bounds over existing works.

**Weaknesses:**

The presentation is at times unclear, leading to a disjointed reading experience.

Additionally, the paper offers limited technical innovation. Most of the algorithmic framework and techniques appear to be adapted from previous works.

**Questions:**

1. The tree mechanism in Algorithm 1 is hard to understand and seems logically inconsistent. Specifically, regarding the function NODE, what is its intended purpose? There appears to be no defined output for NODE. Moreover, in line 13, $k'$ is assigned a value greater than $k$, however, line 14 subsequently tests the condition $k'\le k$, which can never be true. As a result, $S$ remains an empty set and is never updated.

2. In the fourth line of Proposition 2.5, for calculating each $X_i$, should it instead use $\sum_{j=1}^i M_j$  rather than the expression given in the paper  $\sum_{j=1}^i M_i$?

3. In (Cutkoskyetal.,2023), $\Delta_{t+1}$ is updated by $\Delta_{t}+\eta g$ (as stated in their Remark 10), while in Algorithm 2 line 8, $\Delta_{t+1}$ is updated by $\Delta_{t}-\eta g$. Could you clarify the rationale behind this difference?

4. In Theorem 3.1, while the sample complexity has been reduced by a factor of $\Omega(\sqrt{d})$ compared to (Zhang et al., 2024), this comes at the expense of increasing the number of random directions $m$ from $d$ to $d^2$, potentially resulting in a longer runtime.

---

> ### Author Response · Authors · 2024-11-20
> **Response to reviewer hF98**
>
> We thank the reviewer for their time, and appreciate their constructive efforts in helping us strengthen our work.
>
> We have attempted to address the novelty concerns in the common response above. Below we address the other questions raised by the reviewer:
>
> 1,2.: Thank you for pointing out the typos in the tree mechanism section, we appreciate the reviewer’s detailed review. The condition in Algorithm 1 should indeed be $k' \leq t$, and the expression in the fourth line of Proposition 2.5 should be $\sum_{j=1}^i M_j$. We have corrected these errors in the revised version.
>
> The tree mechanism approach ensures that the algorithm maintains strong differential privacy guarantees while minimizing the error introduced by noise. It has become a standard tool in differential privacy and in private optimization, and further details can be found in lecture notes and textbooks. To clarify the tree mechanism, we provide a brief and non-technical explanation:
>
> Suppose we are given a sequence of real numbers $X_1, X_2, \ldots, X_n$, where each $X_i$ lies in $[0, 1]$, and we aim to compute the cumulative sums $\sum_{j=1}^i X_j$ for all $i \in [n]$ while preserving differential privacy.
>
> A naive approach would add independent Gaussian noise to each $X_i$, but this results in an error proportional to $\sqrt{n}$, which grows poorly with $n$. The binary tree mechanism improves on this by organizing the computations into a hierarchical structure:
>
>
> - Binary Tree Construction: We construct a complete binary tree with $n$ leaves, where each leaf corresponds to one $X_i$. Each internal node of the tree represents the sum of the values of its descendant nodes. For instance:  A node spanning the range $(u, v)$ represents $\sum_{j=u}^v X_j$; Its left child spans $(u, m)$, and its right child spans $(m+1, v)$, where $m$ is the midpoint of $[u, v]$.
>
> - Adding Independent Noise: To privatize the sums, independent Gaussian noise is added to the output of every node in the tree. This means that, for any node spanning a range $(u, v)$, the privatized sum is:
> $
>     \text{PrivSum}(u, v) = \sum_{j=u}^v X_j + N_{u,v},
> $
>     where $N_{u,v}$ is Gaussian noise with an appropriate scale determined by the privacy parameters.
>
> - Querying Cumulative Sums: To compute any cumulative sum $\sum_{j=1}^i X_j$, the mechanism selects at most $\log n$ nodes from the tree whose ranges cover $[1, i]$. The noisy outputs from these nodes are combined to produce the privatized result. This hierarchical structure ensures that the total error scales with $O(\log^2 n)$, which is significantly smaller than $\sqrt{n}$ for large $n$.
>
> - Role of $\text{NODE}(t)$: In Algorithm 1, the function $\text{NODE}(t)$ determines the set of nodes needed to cover the range $[1, t]$ in the tree in a greedy way. These nodes are then used to determine which Gaussians should be added to compute the privatized sum.
>
>
>
> We hope this explanation clarifies the tree mechanism and its implementation. Please let us know if you have additional questions or feedback.
>
>
> 3. These are precisely the same update rules in disguise, since the authors therein “subtract” the update whereas we “add” it, so they are defined as negations.
> In detail, in remark 10 therein, they query the gradient at  $z_n:=x_n+(s_n-1)\Delta_n$, and since $s_n\sim[0,1]$, it holds that $(s_n-1)\sim[-1,0]$, or in other words $z_n=x_n-s’_n\Delta_n$ where $s’_n\sim[0,1]$.
>
> 4. This is true. We therefore complement our results by a first-order algorithm (in Appendix C) with a substantially reduced oracle complexity which avoids this blow-up - see Remark C.5, and also our top-comment for further discussion.
>
> We kindly ask the reviewer to consider reevaluating the rating in light of these corrections and clarifications, as they address the key concerns they raised.

---

> > ### Comment · Reviewer_hF98 · 2024-11-23
> > **Re: Response to reviewer hF98**
> >
> > Thank you for the clarification, particularly regarding the explanation of the tree mechanism. I have a follow-up question about its use in Algorithm 3. In each iteration $t$, the algorithm samples a minibatch $S_t$ from the unused data and computes the gradient $g_t$ based solely on $S_t$. Since each data sample is used only once and never revisited, what is the rationale behind employing the tree mechanism, which introduces noise to the cumulative gradient sum rather than to each instantaneous gradient?

---

> > > ### Author Response · Authors · 2024-11-23
> > >
> > > Thank you for your question. If we do not employ the tree mechanism and instead add independent Gaussian noise (denoted as $\zeta_t$) to privatize the gradient $g_t$ at each step, the noise in the cumulative gradient sum would grow as $\sum \zeta_t$, leading to a significant increase in total noise.
> > >
> > > Additionally, if we were to use the true gradient $g_t$ directly when computing the cumulative gradient sum rather than the privatized gradient, we would effectively be reusing the true gradient multiple times. This would require adding larger Gaussian noise to account for composition effects, also impacting the utility of the gradient estimates.
> > >
> > > Hence, the tree mechanism is an ideal choice in scenarios where no data is reused and the cumulative gradient sum is required.

---

> > > > ### Comment · Reviewer_hF98 · 2024-11-23
> > > >
> > > > Thank you for your response and detailed explanation. I would like to clarify further: do you maintain a separate tree mechanism for every $\Sigma$ iterations for Algorithm 3?

---

> > > > > ### Author Response · Authors · 2024-11-23
> > > > >
> > > > > Thank you for your timely response. Yes, we use a separate tree mechanism for every $\Sigma$ iteration. A shared tree mechanism across iterations could risk privacy leakage and compromise the differential privacy guarantee. We will ensure to clarify and emphasize this point in the revised version.

---

> > > > > > ### Comment · Reviewer_hF98 · 2024-11-23
> > > > > >
> > > > > > Thanks for the clarification. I look forward to the revised manuscript

---

> > > > > > > ### Author Response · Authors · 2024-11-24
> > > > > > >
> > > > > > > Thank you for your thoughtful feedback and for taking the time to reassess our work. We look forward to incorporating your suggestions in the revised manuscript. If you have any other questions or comments, please let us know!

---

### Official Review · Reviewer_UGBN · 2024-11-03

**Soundness:** 3
**Presentation:** 4
**Contribution:** 3
**Rating:** 8
**Confidence:** 3

**Summary:**

This paper studies the problem of designing Differentially Private Algorithms for Nonsmooth Nonconvex Optimization. It specifically studies the zeroth order settings. Thanks to more careful analysis, the paper is able to improve on the dimension dependence of the previous results. It also extends the past result to the ERM settings.

**Strengths:**

- The paper is a fairly simple extension of previous results. The authors are able to improve the past sample complexity by an order of $O(\sqrt{d})$ by using a high probability subgaussian bound on the sensitivity of the queries.

- The paper also extends the results to other settings.

- The generalization statement (Proposition 5.1) is a cool result to show the validity of the ERM approach for solving the Goldstein-stationary point.

- The paper is well written overall.

**Weaknesses:**

- The paper is basically using the same algorithm proposed by [1]. This is not a huge issue since they are able to make some nice modifications to improve the sample complexity. However, this does limit the potential impact of the paper.

- I also think $m$ is quite large, which would make it really inefficient to run in practice. Currently, m can be something like $O(d^2T^{4/5})$, which is very hard to do in practice.

- It would be interesting if there were some matching upper bounds.

[1] Zhang, Qinzi, Hoang Tran, and Ashok Cutkosky. "Private zeroth-order nonsmooth nonconvex optimization." arXiv preprint arXiv:2406.19579 (2024).

**Questions:**

- Why did the authors decide to use Gaussian Mechanism instead of the tree mechanism for ERM?

---

> ### Author Response · Authors · 2024-11-20
> **Response to reviewer UGBN**
>
> We thank the reviewer for their time and effort, and are encouraged by their appreciation of our contribution.
>
> Regarding the question the reviewer raised: In the ERM setting, we re-use samples, and therefore must use privacy composition theorems in order to argue about the privacy of the overall algorithm. Since Gaussian composition guarantees are simpler to apply than their counterparts for the tree mechanism, we choose to use the Gaussian mechanism to avoid further unnecessary complications. We thank the reviewer for raising this question, and we will clarify this in the revised version.

---

> > ### Author Response · Authors · 2024-11-26
> >
> > Dear Reviewer,
> >
> > Thank you once again for your detailed comments. We would greatly appreciate knowing if our responses have adequately addressed your concerns and questions, and we would also be happy to engage in any further discussions if needed.

---

### Official Review · Reviewer_KZV5 · 2024-11-04

**Soundness:** 3
**Presentation:** 3
**Contribution:** 2
**Rating:** 6
**Confidence:** 3

**Summary:**

The paper presents algorithms to improve sample complexity in differentially private (DP) nonsmooth, nonconvex (NSNC) optimization. The authors propose two zero-order algorithms that improve the results over Zhang et al. 2024
1. Single-pass, sqrt(d) improvement over Zhang et al. 2024. The authors also establish a dimension-independent “non-private” term, which is not known before for NSNC DP optimization.
2. A multi-pass algorithm further improves the sample complexity, yielding the first algorithm to preform private ERM with sublinear dimension-dependent sample complexity for NSNC objectives.
Additionally, the authors show that Goldstein-stationarity generalizes from the ERM to the population.

**Strengths:**

- The paper is well-structured and easy to follow.
- The results improve over prior state of the art, establishing the first DP NSNC ERM algorithm with sublinear dim-dependent sample complexity. The non-private term is dimension-independent, which improves over the previous dimensional dependent result in Zhang et al. 2024.

**Weaknesses:**

My main concern is about contextualizing the contribution:
Like Zhang et al. 2024, this paper also heavily relies on “Online-to-Non-Convex conversion” (O2NC) of Cutkosky et al. (2023). The authors also mention that a lower bound is unknown, making it hard to assess the contribution beyond incremental improvement.

A discussion of the Tree Mechanism is missing. It would be very hard for readers not familiar with the Tree Mechanism to understand.
Typo: page 2 and in particular is the first algorithm to [preform] private ERM

**Questions:**

1. I am confused by the explanation of the improvement on the non-private term Remark 3.2. The authors explain that
> while the optimal zero-order oracle complexity is d/αβ^3 (Kornowski & Shamir, 2024), and in particular must scale
with the dimension (Duchi et al., 2015), the sample complexity might not.
Since the algorithm is one-pass, then the sample complexity would be worse than the oracle complexity?
2. Is Online-to-Non-Convex conversion optimal? (Related to the weakness above) If not, any algorithms based on it will be suboptimal.

---

> ### Author Response · Authors · 2024-11-20
> **Response to reviewer KZV5**
>
> We thank the reviewer for their time and effort, and are encouraged by their appreciation of our contribution and presentation.
>
> We also thank the reviewer for their questions, and we will modify writing to better explain the points they have raised. We address them as follows:
>
> 1. Indeed, while the single-pass algorithm uses each datapoint $\xi_i$ once throughout its run, in each step it queries $m$ oracle calls in order to stabilize the response (hence reducing sensitivity), as seen for example in line 6 of Algorithm 3. In other words, the oracle complexity is m times the sample complexity, and hence it does not “break” known lower bounds.
>
> 2. Yes, this is a good point: the O2NC itself is optimal, as it is matched by a (non-private) lower bound, as discussed in the original paper by Cutkosky, Mehta and Orbaona. The difficulty in having a matching lower bound lies in the private regime, for which even in the more well-studied smooth non-convex case there are currently gaps between known upper and lower bounds.

---

> > ### Author Response · Authors · 2024-11-26
> >
> > Dear Reviewer,
> >
> > Thank you once again for your detailed comments. We would greatly appreciate knowing if our responses have adequately addressed your concerns and questions, and we would also be happy to engage in any further discussions if needed.

---

> > ### Comment · Reviewer_KZV5 · 2024-11-26
> >
> > Thanks for your response. I will keep my score.

---

### Author Response · Authors · 2024-11-20
**Rebuttal: Common comment**

We thank the reviewers for their time and effort. We are encouraged by their appreciation of our contributions, as well as by their constructive efforts in helping us strengthen our work.
We address all questions by responding directly to each review. Here we briefly reiterate our key responses.

**Novelty**:

Some reviewers asked about the novelty of our results and techniques, compared to previous results.
The sample complexity we derive in this work is not only better than previous results, it was previously erroneously claimed to be impossible - so we believe that even the fact that we have been able to get to these results carries novelty.

Furthermore, our analysis introduces several factors which were not used by previous works on this subject, which were also acknowledged by some reviewers.
First, by identifying a simple concentration argument for the gradient estimator, we reduce its sensitivity, already leading to a significant improvement over prior work. Moreover, to further improve the sample complexity, we analyze the empirical objective which was not considered by any prior work on DP nonsmooth nonconvex optimization, and prove that Goldstein-stationary points generalize from the empirical loss to the population. Not only does this generalization result lead to better sample complexity in the DP setting, we believe it is of independent interest in nonsmooth nonconvex optimization in general.

We would like to add that, in addition to the points above, it is indeed true that our work utilizes several well-studied techniques in DP optimization and in NSNC optimization. We believe that the fact that we are able to improve the sample complexity to the extent that we have (beyond what some thought possible), and the clarity in which we properly acknowledge the use of prior work in order to do so, is an advantage of our writing instead of a drawback. Throughout the paper we explain how we modify and build on top of previous algorithms in order to get these improved results, as generally acknowledged by the reviewers.

**Presentation of the tree mechanism**:

We thank the reviewers that have noted that the presentation of the tree mechanism can be improved, including spotting typos in the index counters.
While this mechanism is standard in the DP literature and our use of it is relatively straightforward, we will properly revise its introduction and add intuition about its utility and corresponding guarantee for readers who are less familiar with it.

**Efficiency**:

Some reviewers pointed out that, as we discuss in the paper, the improvement in sample complexity is at the expense of increased oracle complexity. We would like to emphasize two related points:
- First-order algorithm with significantly better runtime: In Appendix C, we introduce a first-order algorithm with a substantially reduced oracle complexity (see Remark C.5). We believe the first-order analysis complements our zero-order results, and in particular addresses the discussed issue. We will make sure to further emphasize this in the main text, in which we chose to provide only our zero-order algorithms due to lack of space, and as they directly correspond to the previous work on this subject.

- Trading off sample complexity and runtime: A key advantage of our analysis is that it easily allows trading-off sample vs. oracle complexity, which is controlled by the assignment of $m$ in the algorithm. Indeed, on one hand the oracle complexity clearly grows with $m$, while on the other hand the sensitivity bound we derive has an additional term which decays with $m$ (Lemma 3.3). We choose to assign $m$ large enough in order for this additional term to be negligible, hence reducing the sample complexity as small as possible, but this is not generally required by the analysis which enables trading them off smoothly.

---

### Meta-Review · Area_Chair_2UU6 · 2024-12-20

**Metareview:**

This paper presents new single-pass and multi-pass algorithms for differentially private (DP) optimization of nonsmooth nonconvex. Authors provide new state-of-art art sample complexity for both these types of algorithms. Algorithm similar to Zhang et al., 2024, but authors provide a new analysis utilizing the smoothness of the randomly smoothed envelope to improve sensitivity of zeroth order gradient and therefore improve utility trade off of private optimization. However, many reviewers had concerns about presentation and novelty of the techniques used. Paper also do not provide experimental validation of their results.

**Additional Comments On Reviewer Discussion:**

Authors provided more conceptual clarity, but they didn't improve the presentation.

---

### Decision · Program_Chairs · 2025-01-22

Reject